# communications
# engineering

# Integrated stretchable pneumatic strain gauges for electronics-free soft robots

Anastasia Koivikko [1], Vilma Lampinen[1], Mika Pihlajamäki [1], Kyriacos Yiannacou[1], Vipul Sharma[1] & Veikko Sariola [1]✉

In soft robotics, actuators, logic and power systems can be entirely fluidic and electronics-free. However, sensors still typically rely on electric or optical principles. This adds complexity to fluidic soft robots because transducers are needed, and electrical materials have to be incorporated. Herein, we show a highly-stretchable pneumatic strain gauge based on a meandering microchannel in a soft elastomer material thus eliminating the need for an electrical signal in soft robots. Using such pneumatic sensors, we demonstrate an all-pneumatic gripper with integrated pneumatic strain gauges that is capable of autonomous closure and object recognition. The gauges can measure at least up to 300% engineering strains. The sensor exhibits a very stable signal over a 12-hour measurement period with no hysteresis. Using pneumatic sensors, all four major components of robots—actuators, logic, power, and sensors—can be pneumatic, enabling all-fluidic soft robots with proprioception and exteroception.

[1] Faculty of Medicine and Health Technology, Tampere University, Tampere, Finland. ✉email: veikko.sariola@tuni.fi

The four key components of robots are sensors, actuators, logic, and power source. In classic robots, these components are usually electrical. However, in soft robots[1]—robots made out of soft materials—fluidic actuators[2–6] have several advantages over other actuator types[7,8] (e.g., shape memory alloys and cable-driven actuators): they can obtain high grasping forces[9], are fast to actuate[10], and achieve large deformations[11]. Breakthroughs in soft pneumatic actuators[2–4] have inspired the use of fluidics for other soft robot components as well. Pneumatic circuits have been used for controlling soft robots[12–15], and gas from chemical reactions has been used as a pneumatic power source[16]. This progress suggests that sensors in soft robots could also be entirely fluidic. However, soft fluidic sensors have received less attention than the other soft robot components.

One of the most important sensor types for robots is the strain sensor, which in different configurations can measure exteroceptive information (e.g., force, pressure)[17–19] or proprioceptive information (e.g., posture, curvature)[3,20]. Currently, most strain sensors used in soft robots output an electrical signal. Many different sensors have been proposed, including resistive[21], capacitive[22–24], triboelectric[25], and optical[17,26] sensors. One of the simplest electrical sensors is the resistive strain gauge. The resistive strain gauge consists of a meandering resistive path, where the resistance changes with the strain. Since the strains in soft robots can be large (e.g., up to 1000%[27]), alternative methods to classical metal foil strain gauges have been proposed, such as liquid metals[28], printed electronics[3,23], and different ionic conductors[21].

Adding these types of sensors to soft robots tends to increase their complexity: (1) the interface between the robot material and the sensor material is the typical point of failure (delamination, bubbles in liquid alloys, etc.)[3,29]; (2) the sensor materials are fabricated using different processes, adding multiple steps to the robot fabrication process (filling of liquid alloy channels, printing metallic conductors, etc.)[17,28,29]; (3) separate control and power systems for actuation and sensing are needed; and (4) transducers are needed to shift signals from one energy domain to another.

Soft fluidic sensors fabricated using the same materials as the rest of robot would solve all the aforementioned problems. In the literature, several fluidic sensors have been proposed that are based on measuring the pressure inside a sealed air chamber. When the shape of the chamber changes due to elongation, compression, or bending, its volume changes, resulting in a pressure change inside the chamber. For example, Yang et al.[30] proposed integrating a pneumatic chamber into a gripper for measuring contact force and curvature, Choi et al.[31] proposed a three-axis force sensor based on three radially symmetric pneumatic chambers, and Tawk et al.[32] proposed pneumatic sensing chambers for human–machine interfaces. Related to these ideas, Agaoglu et al.[33] proposed a solution with two chambers, one filled with liquid and another with air, with a channel in between. When the liquid chamber deforms, the water–air interface is displaced, which can be measured by using an image-based measurement.

We argue that methods based on sealing fluids inside closed silicone chambers are inherently unstable for long-term measurements, as any leakage will change the sensor output. Leakages can occur not only through faulty connections but also through cracks and ruptures. Silicone elastomers are permeable to gases and many oils, which leads to slow diffusion of fluids in and out of the chamber over long periods of time.

The electrical equivalent of measuring pressure inside a sealed chamber is charging a capacitor and then measuring the voltage over the capacitor, which is related to varying of the capacitance. In electric sensors, this method is not commonly used, because

any leakage current will lead to discharging of the capacitor. Instead, electrical sensors typically rely on constant input of energy. For example, in resistive strain gauges, a small current passes through a meandering conductor, and the voltage drop over the conductor is proportional to its strain. This begs the question: do resistive strain gauges have analogous fluidic components and what are their advantages over methods based on sealed chambers?

Given the familiarity of robotics researchers and developers with electric circuit analysis, many soft robots have been presented with their electrical equivalent circuit[34]: tanks analogous to capacitors, valves to transistors, and narrow channels to resistors. Kusuda et al.[35] used this electrofluidic analogy of a resistor in their u-shaped bending sensor based on a single pneumatic microchannel. They were able to measure the bending of an actuator by measuring the change in the flow rate in the microchannel.

Here, we report that meandering pneumatic microchannels in extremely soft material (Shore 00-50, 100% modulus 83 kPa) can be used as pneumatic resistive strain gauges (Fig. 1a) analogous to electrical resistive strain gauges. We used the electrofluidic analogy (Fig. 1b) to convert the measured pressure inside the microchannels to fluidic resistance and showed that when the channels are elongated, the fluidic resistance increases (Fig. 1c). The microchannels in the pneumatic strain gauges (Fig. 1d, e) are fabricated by casting elastomers into 3D printed or photolithographically defined molds. We show that the strain gauges are stable during long-term loading, show no hysteresis, and withstand up to 300% engineering strains (Fig. 1f). We integrated such strain sensors into a soft robotic gripper and showed that they can be used to measure proprioceptive and exteroceptive information: the sensors can be used to detect the curvature of each gripper finger and they can be used as tactile pressure sensors to detect when the gripper touches an object. Furthermore, by connecting the output from an exteroceptive sensor directly into the driving pressure of a soft fluidic actuator, we realized a gripper that self-closes when coming in contact with an object, entirely without any electronic components. With soft fluidic sensors, the palette of soft fluidic components is complete. Thus, this work contributes to the development of electronics-free, fully pneumatic soft robots with perception.

## Results

To develop a theoretical understanding of soft pneumatic strain gauges, we make the following simplifying assumptions:

1. The fluid is isothermal compressible ideal gas: $P \sim \rho$, where $P$ is the pressure and $\rho$ is the density.
2. Flows are laminar and subsonic.

With such assumptions, it can be shown that the pressure drop in a long channel with a constant cross-section, such as the pneumatic strain gauge, is given by[36]:

$$R\dot{m} = P_{\text{gauge}}^2 - P_{\text{atm}}^2 \qquad (1)$$

where $\dot{m}$ is the mass flow, $P_{\text{gauge}}$ is the absolute pressure at its inlet, $P_{\text{atm}}$ is the absolute pressure at its outlet (atmospheric pressure in our case), and $R$ is the coefficient of proportionality, that is, the pneumatic resistance, which depends on the geometry of the pneumatic strain gauge. Note that we define resistance using the mass flow instead of the volume flow because there is less risk of confusion when the fluid is compressible[37] since the mass flow is conserved from the inlet to the outlet, whereas the volume flow varies with the pressure. The two are related simply by the ratio of density $\rho$. The appearance of second power of pressure is a direct consequence of the fact that $\rho \sim P$ for ideal

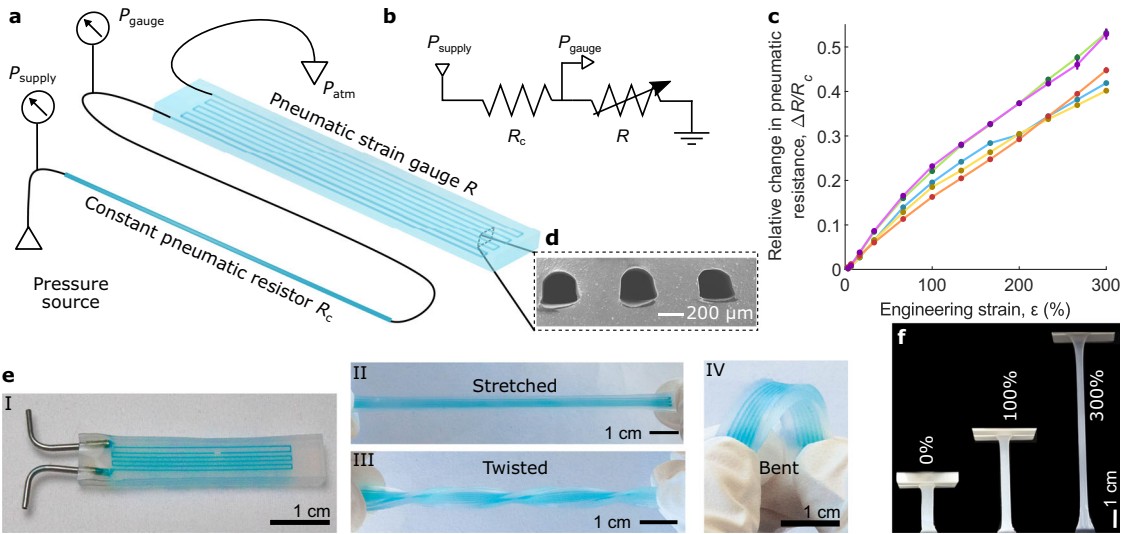

**Fig. 1 Stretchable pneumatic strain gauge. a** Schematic of the soft strain gauge. The measurement setup consists of a constant pneumatic resistor $R_c$ and a pneumatic strain gauge $R$ in series. A constant positive supply pressure $P_{supply}$ is measured and applied to the constant pneumatic resistor $R_c$. The measured pressure between the resistor and the strain gauge is $P_{gauge}$. The strain gauge is vented to atmospheric pressure $P_{atm}$. The pressure drop over the strain gauge depends on its strain. **b** Electric equivalent for the pneumatic circuit in **a**). **c** Relative change in the pneumatic resistance $\Delta R/R_c$ of the strain gauge as a function of the applied engineering strain $\varepsilon$. The five different colored curves are from different samples, and each curve is an average of five measurements with that sample. Error bars (almost too small to be seen) denote standard deviation. **d** A scanning electron micrograph of the channel cross-sections. **e** Photographs of the soft pneumatic strain gauges when they are relaxed (I), stretched (II), twisted (II), and bent (IV). Channels are filled with blue liquid to visualize them. **f** Photographs show the gauge in 0%, 100% and 300% strain.

gasses. This is different from incompressible fluids, where $\rho$ is constant and the volume flow is proportional to the difference in the first power of pressures[38].

When the pneumatic strain gauge is placed in series after a constant pneumatic resistor $R_c$ (Fig. 1a, b), with a pressure $P_{supply}$ supplied at its inlet, the mass flow is the same through both resistors and we have

$$R_c \dot{m} = P_{supply}^2 - P_{gauge}^2 \tag{2}$$

In our measurements, we measure pressures instead of flows. Without flow measurements, we cannot recover mass flows or absolute resistances, but only the ratio of the two resistances. Using Eqs. (1) and (2), we get:

$$\frac{R}{R_c} = \frac{P_{gauge}^2 - P_{atm}^2}{P_{supply}^2 - P_{gauge}^2} \tag{3}$$

We now assume that all pressures are close to the atmospheric pressure, i.e., $P_{gauge,supply} = P_{atm} + \Delta P_{gauge,supply}$, where $\Delta P_{gauge,supply}$ are small pressures relative to the atmospheric pressure. Substituting these in Eq. (3) and dropping the second order terms, we recover the classic linear equation of a voltage divider:

$$\frac{R}{R_c} = \frac{\Delta P_{gauge}}{\Delta P_{supply} - \Delta P_{gauge}} \tag{4}$$

This is identical to the equation for incompressible fluids because when the pressure drops are small, the compressibility effects can be neglected. However, as we will use supply pressures as high as 60 kPa, there is a small but non-negligible difference between values obtained by Eqs. (3) or (4). Since we calibrate how $R$ depends on the strain, in practice it does not matter whether Eqs. (3) or (4) is used, as the sensor calibration corrects the minor non-linearities. Thus, we use Eq. (4) throughout the paper.

When the strain gauge is elongated, its resistance changes, i.e., $R = R_0 + \Delta R$. It is clear that both increasing the channel length and decreasing its cross-sectional area should increase the

resistance, that is, result in a positive $\Delta R$. We found that $R_0$ varied with the sample, possibly because the pneumatic resistance was sensitive to constrictions. However, $\Delta R$ was more consistent across samples, which is why we generally plotted $\Delta R/R_c$ instead of $R/R_c$ throughout the paper.

**Highly stretchable pneumatic strain gauges.** To demonstrate that microfluidic channels can be used as highly stretchable pneumatic strain gauges, we fabricated meandering channels (width and height: 200 μm) in soft silicone elastomer (Ecoflex 00-50), as shown in Fig. 1a, e. The total length of the channel was 183 mm, with five turns, resulting in a sensor with an area of $30 \times 3.84$ mm. The soft resistive strain gauge was denoted as $R$ and was connected in series with a constant pneumatic resistor, called $R_c$ (a narrow tube), and a constant positive pressure $P_{supply}$ of 60 kPa was applied to the circuit. The pneumatic ground was the atmospheric pressure $P_{atm}$ (Fig. 1a). The dimensions of the constant pneumatic resistor were chosen so that initially $P_{gauge}$ was ~10 kPa and rose to ~24 kPa when 300% engineering strain $\varepsilon = \frac{L-L_0}{L_0}$ was applied to the strain gauge, where $L$ is the elongated length of the strain gauge, and $L_0$ is its initial length.

To see the change in the pneumatic resistance of the channels under varying strain, we recorded $\Delta R/R_c$ as a function of the applied engineering strain from 0% to 300%. Figure 1c shows the change in $\Delta R/R_c$ under applied strain for five strain gauge samples. An example of the raw pressure data is shown in Supplementary Fig. 1. Clearly, $\Delta R/R_c$ increased within the applied engineering strain. There was some variation between the samples, but the general behavior was very similar. No sample ruptured or otherwise failed with strains less than 300%; first failures were observed when the strains reached close to 500%. Altogether, these results show that our proposed pneumatic strain gauges can be used to measure extremely large strains.

In Supplementary Fig. 2, we plotted the data from Fig. 1c using the square pressure equations for compressible fluids (Eq. (3)) instead of the small pressure approximation (Eq. (4)). The

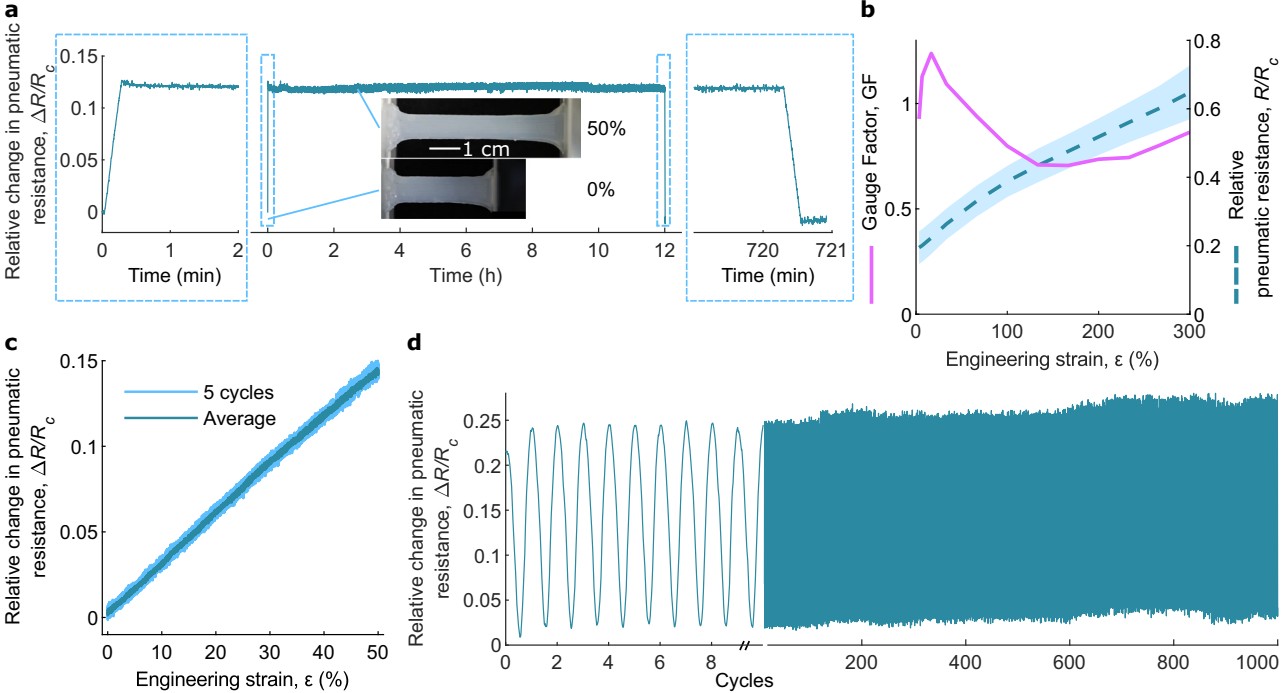

**Fig. 2 Stability and sensitivity of the soft pneumatic strain gauges. a** Stability of the strain gauge when a static 50% engineering strain is held for 12 h, with insets showing photographs from the experiment. **b** Gauge factor of the pneumatic strain gauge. The dashed line is the mean $R/R_c$ of the five tested samples shown in Fig. 1c, with the shaded blue area showing standard deviation. The solid pink line is the gauge factor (average of five strain gauges), calculated using Eq. (6). **c** Hysteresis of the strain gauge when 50% engineering strain is applied for five cycles (strain rate: 0.01 mm s$^{-1}$). The strain gauge shows no hysteretic behavior. **d** Stability of the strain gauge when a cyclic 67% engineering strain is applied for 1000 cycles (frequency: 0.25 Hz).

supplementary figure shows that there is a small difference between Eqs. (3) and (4), but it is very minor. We conclude that the compressibility effects are small in the pressure ranges used.

**Strain gauge stability and sensitivity under tensile strain**. To characterize the long-term stability of the fabricated soft pneumatic strain gauge, we recorded $\Delta R/R_c$ while maintaining a gauge at 50% engineering strain for 12 h (Fig. 2a). The value of $\Delta R/R_c$ drifted from 0.127 to 0.119, with the relative change being 5.9%. After the release, $\Delta R/R_c$ returned almost to the initial value, and the strain gauge did not show notable overshooting behavior. Other two strain gauge samples with same design were also tested (Supplementary Fig. 3), and all three strain gauges had similar stable response over time. Overall, the long-term response of the strain gauge was reliable and reversible.

A leak in the strain gauge could change the strain gauge response. To study this, we measured the strain gauge response while artificially inducing leaks to the strain gauge. In the first experiment, we attached a valve in parallel with the strain gauge and opened the valve to varying degrees. Supplementary Fig. 4a shows the results. When the valve is fully open, the strain gauge signal is fully grounded, and no signal is observed. However, as the valve is partially open, we can always measure some signal, albeit with a smaller signal-to-noise-ratio. This shows that leaks reduce mainly the signal amplitude, but this can be compensated with recalibration. In the second experiment, we punctured the strain gauge in different positions with 0.3 mm and 0.5 mm needles. Supplementary Fig. 4b shows the results. Supplementary Fig. 4b shows that the strain gauge loses some of the signal amplitude as it starts to leak due to punctures, but again, the degradation of the signal is gradual as more punctures are added and at least in the case of minor leaks, this loss of signal can be compensated by recalibrating the strain gauge. Overall, these two

experiments show that leaks do not result in catastrophic failure of the strain gauge, but rather just lower the signal-to-noise-ratio.

The sensitivity of a strain gauge can be studied by its gauge factor (GF), which is the relative change in the output signal to the applied strain:

$$\text{GF} = \frac{dR}{dL}\frac{L}{R} \tag{5}$$

Owing to our measuring configuration, we could not measure $R$, only $R/R_c$. Additionally, we wanted to study the gauge factor as a function of the engineering strain $\varepsilon$ ($dL = L_0 d\varepsilon$). By substituting these variables into Eq. (5), we obtained

$$\text{GF} = \frac{d(R/R_c)}{d\varepsilon} \cdot \frac{R_c}{L_0} \cdot \frac{L}{R} = \frac{d(R/R_c)}{d\varepsilon} \cdot \frac{1+\varepsilon}{R/R_c} \tag{6}$$

which is shown in Fig. 2b. The GF is calculated by using an average $R/R_c$ of the five strain gauge samples (dashed blue line in Fig. 2b). The GF varied from 0.71 to 1.2 but was very small in general. Doubling the length roughly doubled the resistance.

The rate-independent hysteretic behavior of the strain gauge was studied by recording $\Delta R/R_c$ when 50% engineering strain was applied for five cycles. In each cycle, the strain gauge was first stretched and then returned to its original length, at a constant strain rate. The strain rate was kept low (0.01 mm s$^{-1}$) to avoid rate-dependent effects, e.g., dynamic viscoelastic effects. The results are shown in Fig. 2c. The rate-independent hysteresis, if there was any, was smaller than the noise level of the sensor (5% with the tested range). Even after averaging the data from the five cycles, no hysteresis was observed (Fig. 2c). Two other strain gauge samples with the same design were also tested and they showed similar non-hysteretic behavior (Supplementary Fig. 5).

To study the rate-dependent hysteresis (e.g., viscous effects), we did the hysteresis experiment with three different speeds ranging from 0.1 mm s$^{-1}$ up to 1 mm s$^{-1}$ (Supplementary Fig. 6).

No hysteresis could be observed even at the highest speed. However, it should be noted that the pneumatic resistance data and the distance data are from two different devices and had to be manually aligned. Thus, the results only show that the approach and retraction curves have similar shapes, but they cannot be used to conclude if there is some phase lag between the mechanical tester and the sensor output.

To investigate whether the strain gauge is stable under dynamic strain, we recorded $\Delta R/R_c$ when 67% engineering strain was applied for 1000 cycles with a frequency of 0.25 Hz (Fig. 2d). Some minor variations were observed from one cycle to another (Fig. 2d), but overall, the response was very stable for the duration of the whole experiment, which was over an hour. To confirm that these results were reproducible, we repeated the cyclic loading experiment with two additional samples. The results are shown in Supplementary Fig. 7 and are very close to the results in Fig. 2d.

To test the bandwidth of our strain gauges, we repeated the 67% cyclic strain experiments with frequencies ranging from 0.1 up to 1 Hz, which was the maximum frequency our characterization setup could reliably produce. The results are shown in Supplementary Fig. 8. The sensor has a clear response even at 1 Hz, which shows that the bandwidth of the sensor is at least 1 Hz, if not more. Taking the results in Fig. 2d and Supplementary Figs. 7 and 8 together, we conclude that the strain gauge has a stable response even in dynamic strain.

**Strain gauges as curvature sensors in soft actuators.** Soft pneumatic bending actuators are one very common type of soft robotic actuator[10]. To see if our sensor could measure the bending of such an actuator, we integrated a sensor to the bottom of the actuator (Fig. 3a). Both the actuator and the strain gauge were made of the same silicone elastomer (Dragon skin 30, Shore hardness A 30, 100% Modulus 593 kPa). The strain gauge was integrated to the actuator by casting it under the bottom part of the actuator (Fig. 3a). Bending the actuator compressed the strain gauge, decreasing $\Delta R/R_c$.

We characterized the strain gauge in the actuator by measuring the curvature $\kappa$ and $\Delta R/R_c$ while cycling the pressure inside the actuator for 30 cycles. The curvature was measured using machine vision. Figure 3b shows the results. At the initial stage, $\Delta R/R_c = -0.002$ and $\kappa = 3.5\ \mathrm{m}^{-1}$, and at the maximum bending, $\Delta R/R_c = -0.022$ and $\kappa = 24\ \mathrm{m}^{-1}$. The data show that the strain gauge detected when the actuator was bent; however, the signal-to-noise ratio was lower, suggesting that the strains were considerably smaller than the strains tested in Figs. 1 and 2.

To estimate the strain of the gauge when it was bent, we used the ideal bending relationship

$$\triangle \varepsilon = d \triangle \kappa \qquad (7)$$

where $d$ is the distance from the neutral plane. We assumed that the neutral plane was in the middle of the strain-limiting layer (fiberglass), thus $d \approx 2\ \mathrm{mm}$. From Fig. 3b, we obtained $\Delta\kappa = 20\ \mathrm{m}^{-1}$. Using these values, Eq. (7) gives $\Delta\varepsilon \approx 4\%$. At the same time, the relative change in pneumatic resistance was 0.02. These values roughly corresponded to the values in Figs. 1c and 2b, so we concluded that 1) ideal bending relationship can be used for a rough estimate of the actual strain, and 2) the strains during bending are rather small, explaining why the data are relatively noisier compared to the data in Fig. 2.

The data collected up to this point did not preclude the possibility that the sensor was actually responding to the pressure change inside the actuator instead of the bending of the actuator. To confirm that the sensor was measuring the bending, we measured the pressure inside the actuator $P_\mathrm{act}$, $\Delta R/R_c$, and curvature $\kappa$ in four different states: neutral, driven, blocked, and forced. In the neutral state, the actuator was free to bend and no pressure was applied inside the actuator (Fig. 3c, I). In the driven

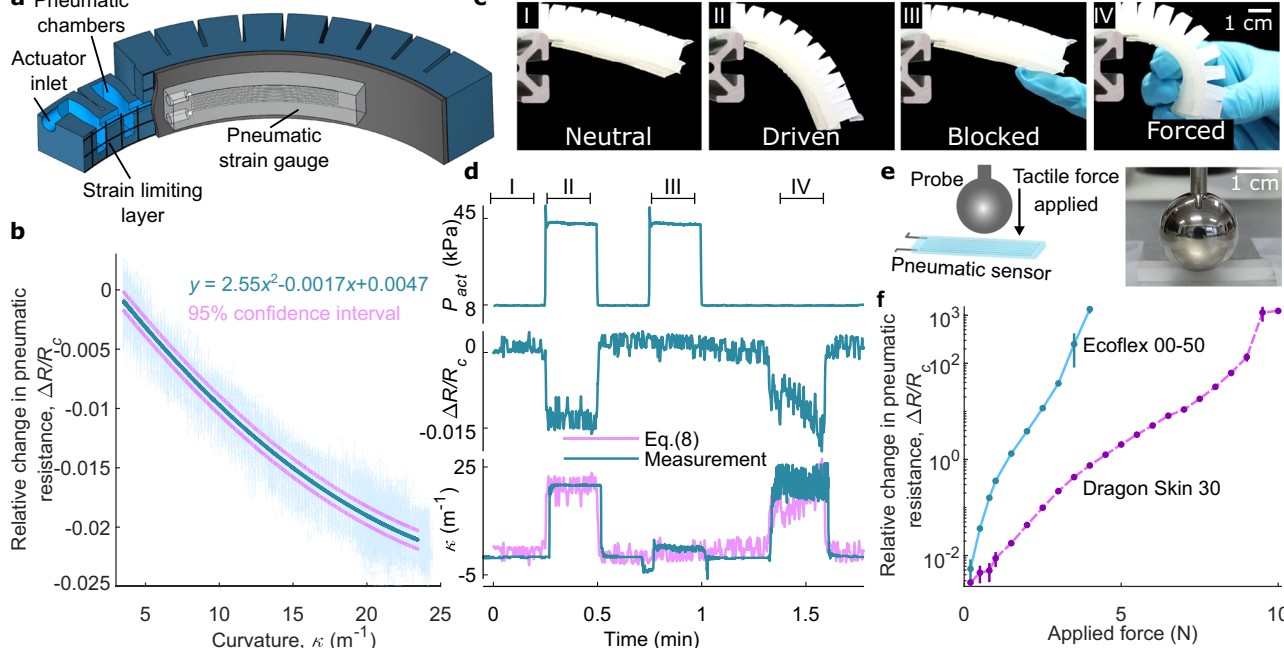

**Fig. 3 The soft pneumatic strain gauge as a curvature and a force sensor. a** Schematic of the soft pneumatic actuator with an integrated strain gauge. **b** $\Delta R/R_c$ as a function of curvature $\kappa$ of the actuator, when the actuator is bent 30 times. **c, d** Decoupling the actuator pressure from the measured curvature. Actuator pressure $P_\mathrm{act}$, relative change in pneumatic resistance $\Delta R/R_c$, and curvature $\kappa$ versus time in four different states: neutral (I), driven (II), blocked (III), and forced (IV). The estimated curvature is calculated using Eq. (8). **e** Schematic and photograph of the compression force measurement. **f** Response of $\Delta R/R_c$ to applied compression force with two different strain gauges: Ecoflex 00-50 (blue line) and Dragon Skin 30 (dashed pink line). Each curve is an average of five measurements with the sample and error bars denote standard deviation.

state, the actuator was free to bend and pressure was applied inside the actuator (Fig. 3c, II). In the blocked state, the actuator was blocked from bending and pressure was applied inside the actuator (Fig. 3c, III). In the forced state, the actuator was forcefully bent by hand, but no pressure was applied inside the actuator (Fig. 3c, IV). The resulting data are shown in Fig. 3d. In the forced state (IV), the curvature data became very noisy, because the manual bending made it more difficult for the machine vision to capture the arc of the actuator. The gauge signal $\Delta R/R_c$ followed the curvature signal more than the pressure signal, indicating that the strain gauge was really sensing the curvature and not just measuring the pressure inside the actuator. We also did linear decoupling using the method of least-squares

$$\kappa_{\text{estimate}} = \alpha(\Delta R/R_c) + \beta P_{\text{act}} + \gamma \qquad (8)$$

where $\alpha = -1249.3\,\text{m}^{-1}$, $\beta = 0.0116\,\text{m}^{-1}\text{kPa}^{-1}$, and $\gamma = 2.66\,\text{m}^{-1}$. Thus, only $\left|\frac{\beta \Delta P_{\text{act}}}{\alpha \Delta R_c}\right| \approx 3\%$ of the strain gauge signal comes from the pressure change; the rest of the signal comes from the actual bending. We conclude that the sensor primarily measures the curvature of the actuator and not the pressure inside it.

**Strain gauges as tactile pressure sensors**. The strain gauges are expected to respond not only to tensile strain but also to transverse compression. The compression decreases the channel cross-section, increasing the pneumatic resistance. The strain gauge was compressed with a spherical probe (diameter 19.05 mm) from the center of the sensor (Fig. 3e). We recorded the change in relative pneumatic resistance under varying compressive force for strain gauges made of Ecoflex 00-50 and Dragon Skin 30. The results are shown in Fig. 3f. Both sensors show comparable increase in $\Delta R/R_c$ although the softer sensors respond to smaller forces as expected. The responses of both sensors are highly super linear and depend on the applied force (notice the logarithmic scale on the y-axis in Fig. 3f).

**Soft pneumatic gripper with proprioception**. To demonstrate proprioceptive soft robots using the proposed strain gauges, we built a three-fingered soft pneumatic gripper with a strain gauge integrated into each finger (Fig. 4a). The electric equivalent circuit for the gripper fingers is shown in Fig. 4b. The pneumatic actuators consisted of expanding pneumatic chambers attached to a flat silicone layer including a glass fiber strain-limiting layer ($C_{1,2,3}$ in Fig. 4b), and they were connected to a 3D printed part through which all-pneumatic connections were made to the actuators and sensors. The gripper was attached to a vertical motor using a screw. All three fingers were actuated with the same pressure signal, so they opened and closed in unison. In this experiment, the driving signal was generated by an external pressure controller. Each finger included an independent strain gauge ($R_{1,2,3}$ in Fig. 4b), which was in series with the corresponding constant pneumatic restrictor ($R_{c1,c2,c3}$ in Fig. 4b) and connected to the common constant pressure source $P_{\text{supply1}}$.

When the gripper fingers were closed, the curvature of the fingers depended on the shape of the target object. The shape and size of the gripped object allowed the fingers to bend by different amounts, and this difference can be observed from the data. To determine whether we could infer the type of the object from the sensor signals alone, we selected three differently shaped objects, a lemon (183 g), a lime (91 g), and a water bottle (24 g), as shown in Supplementary Video 1 and Fig. 4c. Each object was gripped ten times, and data were captured from each strain gauge. The object was changed after every two picking experiments to avoid systemic drift and carry over effects. From each experiment, we calculated the change in sensor signal before and after gripping the object. Figure 4d shows a 3D scatter plot with each axis

corresponding to a different sensor. The data from the water bottle picking is the most scattered. Snapshots of videos recorded from the picking experiments (Supplementary Fig. 9) show that the fingers bend slightly differently when picking the water bottle, because of its odd shape, while there are no noticeable differences in how the lime and lemon are grasped, because their shape is more regular. Thus, these results are strongly indicative that the variations in the sensor signals are in large part due to real variations how the gripper fingers bend during gripping. In conclusion, the objects formed clear clusters, suggesting that the sensor data can be used to infer information about the type of the gripped object.

**Soft pneumatic gripper with exteroception**. In the previous examples, electric pressure sensors and pneumatic controllers were used to facilitate data collection. With pneumatic sensors, it should be possible to create fully pneumatic soft robots. In future intelligent robots, the sensor signals would be presumably routed through logic (the "brain" of the robot), which would then be responsible for controlling the actuators. However, as demonstrated in simple animals such as jellyfish[39], a central brain is not always necessary. Useful behavior can be achieved by routing sensory nerves directly to the muscles.

To demonstrate routing sensor signals directly to the actuators of a soft robot, we placed one strain gauge (channel width and height: 500 µm) onto the gripper palm (Fig. 4a) and connected that palm sensor ($R_4$ in Fig. 4b) in parallel with the pneumatic fingers ($C_{1,2,3}$ in Fig. 4b). When $R_4$ increases, the pressure $P_4$ inside the fingers is to increase. The response of the pneumatic gripper to external compression of the strain sensor was studied by pressing the strain gauge with a solid object (Fig. 4e and Supplementary Video 2). The results show that the compression indeed increases $P_4$ (Supplementary Fig. 10), causing the fingers to bend. When the external compression is released, the fingers return to their initial shape. The closing of the fingers takes ~8 s, which is caused by the pneumatic resistance and capacitance in the gripper fingers, sensor $R_4$ and tubing. The self-closing gripper can be understood as an RC-circuit, where the sensor resistance affects $R$, and the capacitance $C$ is from the pneumatic chambers of the actuators and from the tubing. The time constant $\tau$ of an RC-circuit is given by $\tau = RC$, which suggests that the response time of the gripper can be adjusted by changing the resistance, i.e., the length and cross-section of the meandering strain gauge channel. This was also showed experimentally by using a strain gauge with a smaller microchannel (channel width and height: 200 µm). It then took ~1 min for the gripper to close. To conclude, this is a simple example of an all-pneumatic system that responds to external stimulus by changing its shape, and the response can be tailored to the needs of the application by changing the sensor design. Taking these results together with the results in Fig. 3e, f, the proposed sensors can be used as soft tactile pressure sensors for measuring the external forces in soft robot applications.

## Discussion

We fabricated and characterized a soft pneumatic strain gauge and demonstrated that it can be integrated into soft robots. Supplementary Table 1 compares our results to the different sensors reported in the literature, in terms of their materials, fabrication method, maximum strain, hysteresis, gauge factor, and maximum frequencies tested.

Our sensors are made of soft silicone elastomer, the same material that is commonly used to make the bodies of soft robots, so it can be fabricated using similar materials and processes as the rest of the robot. The sensors do not notably increase the amount

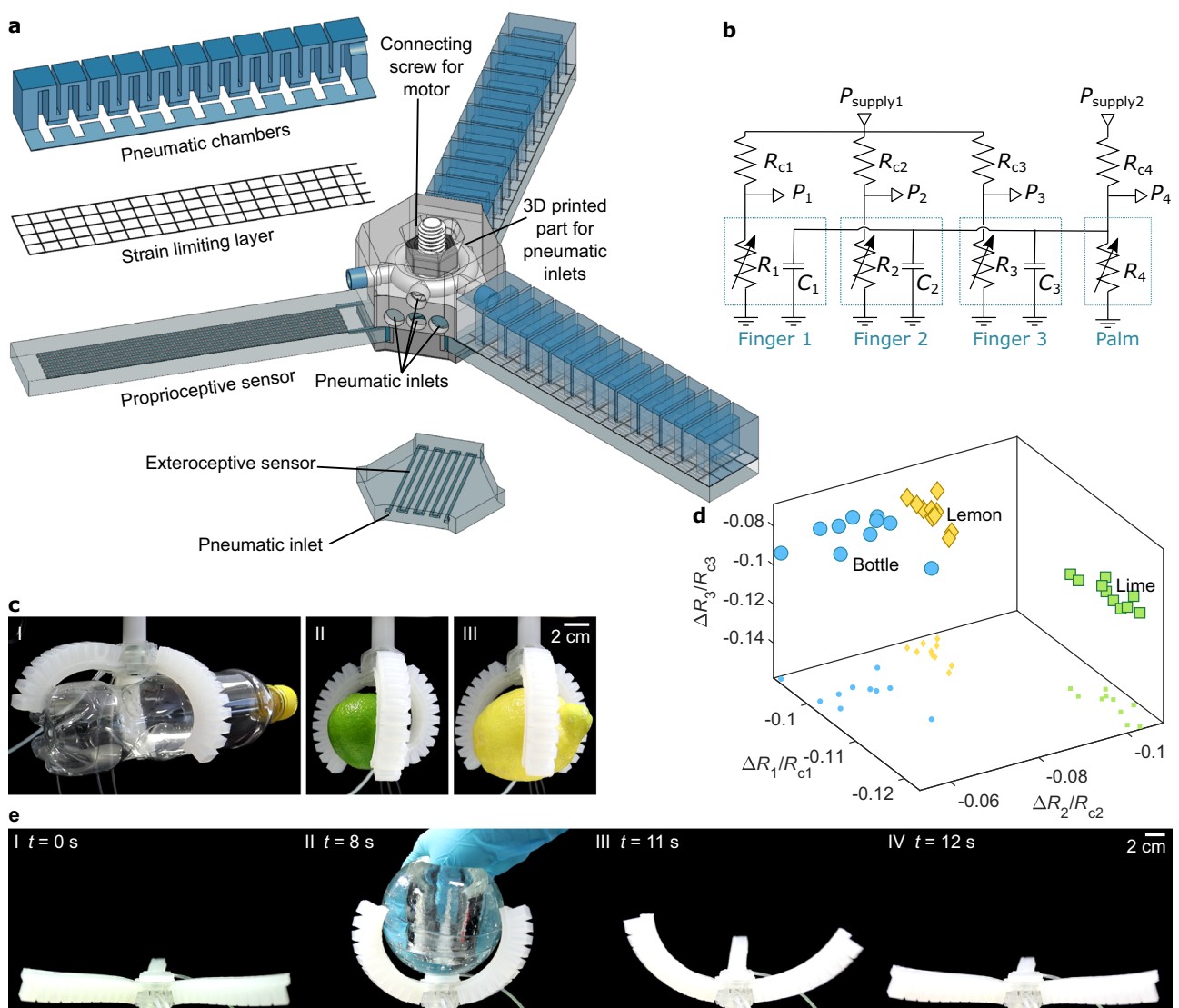

**Fig. 4 Fully pneumatic soft robot with perception. a** Schematic of pneumatic gripper with perception: fingers including proprioceptive sensors and the palm includes an exteroceptive sensor. **b** Electrical equivalent circuit for the gripper: actuators can be considered as capacitors $C_{1,2,3}$, constant pneumatic restrictors as resistors $R_{c1,c2,c3,c4}$, and pneumatic strain gauges as variable resistors $R_{1,2,3,4}$. **c** Photographs of gripped objects: a water bottle (I), a lime (II) and a lemon (III). **d** $\Delta R/R_c$ for each pneumatic strain gauge when the three different objects are each gripped ten times. **e** Movement of gripper fingers when external compression is applied on the palm and then released: gripper fully open (I), fully closed (II), opening (III) and fully open again (IV).

of material consumed in making a pneumatically actuated soft robot, as the sensor is notably smaller than typical pneumatic network actuators. Pneumatic strain gauges do not require any other materials, such as ionic conductors[21], liquid metals[28], or inks[3,23], which is a clear advantage of the proposed approach (Supplementary Table 1). Apart from air and silicone interfaces, heterogeneous materials are not used in our sensors. This is beneficial since delamination often occurs at heterogeneous material interfaces.

Our soft strain gauges were fabricated using 3D printing and soft lithography, which are well-known techniques and have been extensively studied (Supplementary Table 1). Despite this, the fabrication and integration of the strain gauges included several different steps and hours of manual work. Direct 3D printing of entire soft robots has been proposed[16,40]. Our sensors are likely to be also directly 3D printable, provided that the resolution of the 3D printer is sufficient to make the meandering channel. Regardless, we argue that 3D printing is more suitable for prototyping than mass manufacturing. Mass manufacturing of soft

robots with integrated sensing and logic still requires more research and development.

The sensors can be used with up to at least 300% strains. According to datasheets, the silicone elastomer we used breaks at 980% strain, but these values are for bulk material. We observed failures at around ~500% strains, which we attribute to the presence of microstructures in our sensors, where cracks may initiate. Soft pneumatic actuators[4,10] and soft pneumatic logic[12–14] also contain similar microstructures, so these sensors are expected to be just as stretchable as the rest of the soft robot. The stretchability of previously reported electric strain sensors varies greatly, from 2% up to 1400% (Supplementary Table 1)[41]. The 300% strain we reached compares quite favorably to hyper-stretchable electric strain sensors. The principles to increase stretchability and to prevent cracks are well understood: avoid sharp corners, reduce bubbles and pores, and improve adhesion between layers.

We did not observe any hysteresis in our sensors. If there was hysteresis, it was smaller than the noise level (~5% in tested

range). This differs from soft resistive strain gauges, where hysteresis can be over 10% (Supplementary Table 1). The non-hysteretic behavior of our pneumatic strain gauge is more comparable to capacitive[24] and optical[17] strain sensors (Supplementary Table 1).

The gauge factor of our sensor was rather low (~0.7–1), which makes it more suitable for measuring large strains. This is unlike classical electrical piezoresistive strain gauges, where the gauge factor can be as high as ~100,000[42] (Supplementary Table 1). However, the noise level of our sensor was still small enough that we could measure the curvature of a bending actuator, where the strains were rather small (~4%). In the future, the sensitivity of the strain gauge can be increased by using pneumatic amplifiers or transistors, for which their electrical equivalents have previously been used[43]. Many different types of strain sensors[33,44] for measuring small strains already exist, but there are fewer sensors that can measure over 200% strains[41]. The ability of our sensor to measure large strains (at least up to 300%) is potentially useful in motion tracking applications[45] and in soft actuators with linear elongations[11,46].

Owing to limitations of our characterization setup, we were only able to test up 1 Hz frequencies in dynamic testing, and the sensor showed no obvious loss of signal with such a frequency. Supplementary Table 1 shows that soft sensors are not generally tested with frequencies exceeding this. Thus, it is too early to conclude how the bandwidth of our sensor compares to existing sensors. Future work should focus on characterizing all different types of sensors in high-frequency dynamic measurements, to see if there are differences in the maximum bandwidths that can be achieved using different measurement principles.

Drift in strain gauge output is a common challenge during long measurement periods. Resistive strain gauges are temperature sensitive[44], which is often compensated by placing them in Wheatstone bridge circuits. Resistive strain gauges also show larger overshooting behavior than capacitive or optic strain sensors[41]. Pneumatic strain sensors based on measuring the pressure in a sealed chamber[30–32] can also suffer from drift due to the slow diffusion of fluids through the elastomer. Our proposed strain gauge showed stable and reversible behavior during a static 12-h measurement and during repetitive dynamic loading and showed no notable overshooting behavior.

We demonstrated that a soft robotic gripper with integrated pneumatic strain gauges could detect the type of the object being gripped. The results are comparable to the results of Homberg et al.[20], who used resistive curvature sensors in a four fingered gripper to recognize gripped objects. We also showed that the pneumatic strain gauge can be used as a tactile sensor to detected to external forces applied to the gripper. In sum, these results show that pneumatic strain gauges can be used as an alternative to electrical sensors in practical robotic applications. All-pneumatic soft robots could be used in applications where electric devices are prohibited, for example inside magnetic resonance imaging devices or in explosive environments.

We have shown that our sensors respond not only to linear strains but also to transverse compression and bending. In soft robotic applications, the strain gauge could respond to multiple types of stimuli, including body deformations (proprioception) and physical contact (exteroception), and thus the contribution of different stimuli needs to be decoupled. All solutions to this problem are based on the fact that sensors placed in different locations—at the surface of the robot vs. buried deep inside the robot—respond in different degrees to body deformations and to physical contact. Thus, with multiple sensors, placed at different locations, different stimuli can be tested and their contribution to different sensor signals observed. This gives a mapping from the different stimuli to the sensor signals; the mapping can then be inverted to decouple the different stimuli from the observed sensor signals. In the literature, signals from different types of sensors have been decoupled using a linear mapping[17], but also machine learning algorithms have been used[47].

The Hagen–Poiseuille equation for a narrow channel would predict that $R \sim L/A^2$, where $A$ is the cross-sectional area of the channel. Assuming the elastomer is approximately incompressible and the channel deforms evenly, $LA$ = constant, we would expect $R \sim L^3$, that is to say, we would expect the gauge factor to be close to three. However, we found the gauge factor to be closer to one, perhaps even slightly less than one (Figs. 1c and 2b), and this was consistent across the five samples we characterized. This suggests that the flows and deformations inside the channels are more complicated than a simple straining of incompressible rubber and laminar flow model would predict.

The fluid in the strain gauges was air in this study. Using liquids instead could be desirable in many applications, such as underwater soft robots, where liquids are used for their actuators, logic, and power[6,48]. Liquids can usually be assumed incompressible, thus simplifying theoretical analyses. However, pneumatics is particularly attractive for land-based autonomous soft robots due to the availability of air in abundance[49] and because pneumatic power can be generated from chemical reactions[16]. This suggests that our sensors are more likely to see applications in land-based soft robots.

To conclude, the proposed soft pneumatic strain gauges offer a way to integrate proprioceptive and exteroceptive sensing into pneumatic soft robots. Compared to the well-studied liquid alloy-based resistive sensors, our pneumatic sensors have less fabrication steps, as both sensor types require a microfluidic channel, but our sensor does not need the channel to be filled with anything else than air. Compared to electrical resistive sensors, our pneumatic sensors have much less hysteresis and drift. Our sensors also operate on the same pneumatic principles as the soft pneumatic actuators so only one power system is needed. In future, we expect our sensors to see more practical applications in fully pneumatic, electronics-free soft robotic devices.

## Methods

**Soft strain gauge fabrication**. The soft pneumatic strain gauges were made of soft silicone elastomers (Ecoflex 00-50, Shore hardness 00-50 and Dragon Skin 30, Shore Hardness A 30, Smooth-On) by casting them into molds. A mold was made with photolithography (negative photoresist SU-8 3050, Microchem). The mold was treated with trichloro(1H,1H,2H,2H-perfluorooctyl)silane to ease the mold removal. Then the silicone elastomer was poured into the mold, let cure, and demolded. To create closed channels, the fabricated channel structures were bonded to a flat elastomer piece. The bonding was done by spin coating a thin layer of uncured silicone elastomer on the flat elastomer piece, serving as glue between the two parts. After curing the glue, the closed channels were cut to shape. Finally, a needle was used to punch holes for the inlet and the outlet. Connections to the inlet and outlet were sealed with silicon glue (Sil-Poxy, Smooth-On). Supplementary Fig. 11 shows the detailed fabrication steps for the soft pneumatic strain gauges.

**Strain measurements**. The strain gauge was linked to the pneumatic circuit, which included a constant pneumatic resistor $R_c$ (Teflon tube, inner diameter = 0.2 mm) and pressure sensors (015PDAA5, Honeywell, 15 PSI Differential 5 V) for measuring the supply pressure $P_{supply}$ and pressure inside the pneumatic strain gauge $P_{gauge}$. The pressure sensor data were recorded with a data acquisition device (USB-6356, National Instruments). For tensile tests, the soft pneumatic strain gauge was clamped from its ends to a mechanical tester (TA.XT Plus, Stable Micro Systems, Fig. 1f). For the compression tests, the strain gauge was placed below the spherical probe (TA-18A, 3/4" diameter ball, Stable Micro Systems, Fig. 3e). The overall measurement setup in shown in Supplementary Fig. 12a.

**Soft actuator fabrication and strain gauge integration**. The design and fabrication steps of the pneumatic actuator were adapted from the Soft Robotics Toolkit[50]. The dimensions of the actuator were slightly changed: the actuator chambers had a length of 77 mm, a width of 15 mm wide, and a height of 12 mm. Detailed drawings of the actuators are provided in the supplementary material (Supplementary Fig. 13). We cast soft silicone (Dragon skin 30, Smooth-On) to 3D

printed molds made out of polylactic acid (PLA, Prusa I3 MK3, Prusa Research). The actuator was molded in two parts: an upper part including the pneumatic chambers and a lower part including the strain-limiting layer (a glass fiber net). The parts were bonded together by spreading a thin layer of the same silicone between the parts. An air inlet was added to the actuator by first using a biopsy punch and then attaching a tube to the punched hole. The pneumatic strain gauge was glued the bottom of the actuator with a thin layer of silicone.

**Pneumatic control**. The pneumatic control of the actuator was implemented by using a pneumatic control board from the Soft Robotics Toolkit[50]. The solenoid valves were driven by Arduino microcontroller (Arduino MEGA 2560) using pulse-width modulation (56 Hz). The pressure was controlled by changing the duty cycles of the solenoid valves (VQ11OU-5M, SMC). Arduino was controlled by a custom-made MATLAB script. A pressure sensor (100PGAA5, Honeywell, 100 PSI GAGE 5 V) was used to measure the pressure inside the pneumatic actuator.

**Curvature measurement**. For the curvature measurements, the actuator was placed on an aluminum frame from the air inlet side next to a scale bar in front of a black background. A digital camera (Canon, EOS 5D Mark IV) was placed to capture video of the bending actuator. The video was analyzed by a custom-made MATLAB script to find the best fitted circular arc on the bottom of the actuator. The curvature $\kappa = 1/r$, where $r$ is the radius of the actuator, was calculated for each frame.

**Gripper fabrication**. The gripper was fabricated by connecting three pneumatic actuators together with a 3D printed piece (Clear resin, Form 2, Formlabs). Detailed drawings are provided in the supplementary material. The actuator design was same as used previously in the work, except here the strain gauges were integrated in the bottom part of the actuators (Fig. 4a and Supplementary Fig. 13). The strain gauge $R_4$ used for exteroceptive measurements in the gripper palm had a channel with length of 221 mm and a width and a height of 500 μm. The channel cross-section was increased to reduce the pneumatic resistance of the palm sensor, to make the gripper close faster. The palm sensor was attached to the 3D printed connecting piece with a tape. The pneumatic connectors to the actuators and strain gauges were provided through the 3D printed connecting piece (Supplementary Fig. 14).

**Gripper demonstrations**. The actuators of the gripper ($C_{1,2,3}$) were connected in parallel with the palm sensor $R_4$. They were all connected in series with the constant pneumatic restrictor $R_{c4}$ (Teflon tube, diameter 0.5 mm) which was connected to the constant pneumatic source $P_{supply2}$. The strain sensors in the fingers of the gripper were all connected to the constant restrictors ($R_{c1,c2,c3}$) and then to constant pneumatic source $P_{supply1}$.

In the object recognition experiment, the 3D printed connecting piece was attached to the vertical motor (TA.XT Plus, Stable Micro Systems) with a screw (Fig. 4c photographs and a full measurement setup in Supplementary Fig. 12b). The pressure was measured as described in the section Strain measurements. The picked objects were placed in the same position under the gripper manually, and the gripper was moved in contact with the vertical motor (Supplementary Video 1).

In the self-closing gripper experiment, the gripper was placed with fingers pointing up (Fig. 4e and Supplementary Video 2). Then a heavy object was placed onto the palm sensor ($R_4$) by hand, causing the gripper fingers to close. Removing the object caused the fingers to open again.

## Data availability
All the raw data are available from Zenodo (https://doi.org/10.5281/zenodo.6545728)[51].

## Code availability
Scripts to analyze the data are available from Zenodo (https://doi.org/10.5281/zenodo.6545728)[51].

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

## Acknowledgements

We thank LetPub (www.letpub.com) for its linguistic assistance during the preparation of this manuscript. The work was supported by the Academy of Finland, grant numbers #299087, #328265, #311415, #331368, #343408, The Finnish Science Foundation for Technology and Economics, grant number #20190090, and The Finnish Foundation for Technology Promotion, grant number #7528.

## Author contributions

V. Sariola conceived the idea for the research. A.K., V.L, M.P., K.Y., and V. Sharma planned and conducted the experiments. V. Sariola supervised the study. All authors discussed the results. A.K. and V. Sariola wrote the manuscript.

## Competing interests

The authors declare no competing interests.
