## [Peer Review File · Communications Engineering]

Reviewers' comments:

Reviewer #1 (Remarks to the Author):

The paper claims on the development of a pneumatic strain sensor that can be easily fabricated and embedded into soft-bodied robots (hand). This paper is in line with recent attempt on electric-free designs for efficient implementation of soft robots with less burden from rigid components. Other evaluations on the operation of the sensor, and its application on equipping a soft robot hand with proprioceptive sensing were conducted.

The good point in this design is that by only measuring the pressure, the strain or contact information can be extracted. Thus, no need to put anything into the soft body of the robot. However, the most unconvincing point is that it still needs pressure sensors for estimation of the strain based on equations 3-4. This reviewer expected something more than that, like pneumatic sensing+logic+actuation as a loop. With the current design, this reviewer thinks that the bandwidth of the strain sensor cannot be so high. In fact, more evaluation on dynamic change needs to be conducted, and compared to other soft sensors, such as EGain-based ones. The authors mentioned about low hysteresis of the sensor, but it might be due to the fact that it was not conducted at high frequency.

Reviewer #3 (Remarks to the Author):

In this paper, the authors have created soft elastomer material with included microchannels. This fluidic sensing design is used in combination with a pneumatic setup to measure change in relative pneumatic pressure drops resulting in strain measurements.

Though this is an interesting concept. However, there are a number of concerns:

1) The contribution of this paper is not clear, and the argumentation is not logical:

- In the introduction, the authors made a series of conclusions that are vague and, hence, incorrect. For example, it is stated that soft material robots "are simple to fabricate using elastomer casting, are fast, and achieve large deformations..." In my opinion, these statements would need to be revised and differentiated. For instance, there is increasing complexity in some soft robots, e.g., systems that have multiple molding steps. What does the author mean that soft robots are "fast"? Do they mean that the response is quick? This behavior is due to choosing appropriate pressure regulators, for instance.

- The review sections list a number of research efforts without describing the fundamental concepts. For instance, when looking into soft fluidic sensors, there is a common theme among these types of sensors. In fact, the authors conclude that these "methods based on sealing fluids inside a closed silicone chamber". Further, it is "argue[d]" that these methods "are inherently unstable for long-term measurements". Looking at the design and working principle of the authors' work, this fundamental concept has also been adapted here measuring relative pressures. The main critic is that leakages might change the sensor output. If this is the disadvantage of current systems, how does this work overcome this limitation. More important, it would be good to understand how the sensor is affected by leakages.

- The authors refer to the analogy to electric circuits. This analogy is well known. I am not sure what

the reasoning for including this section is?

2) Some sections could benefit from an improved structure. For instance, the Results section begins with proof of concept, then reports on the results when integrating the sensor into a gripper. Later in the section, the authors report on stability and sensitivity. I advise to give a FULL characterization of the strain gauge sensor before reporting on performance criteria when embedding the sensor into a robotic gripper.

3) The unstructured approach is also reflected in Figs 1 and 2. Both figures partially include characterization results of the sensor. I suggest changing these figures. One figure could include the entire overview, setup and results of the main concept. The other figure could then give an overview of the application and its results.

4) It is important to include figures that help to understand the work and that add scientific value. For instance, Fig. 1a does not add significant content. A setup of the real system is missing.

5) I suggest focussing the paper on the actual sensor and the characterization. The application of the sensor to a gripper, which includes a description of the manufacturing process, experiments etc., lacks of depth (due to space limitation).

6) The Discussions section is vague. For instance, the authors respectively claim that the sensor is inexpensive, simple and “trivial” to integrate. There are no proofs for these claims in the paper. The section does not reflect on the contributions. In fact, the authors finally conclude that “This paves the way toward fully pneumatic, electronics-free soft robots” – the title of the paper suggests that this has already been achieved in the paper.

Reviewer #4 (Remarks to the Author):

The authors present a study on pneumatic strain sensors based on meandering microchannels embedded in a soft elastomer material. The sensors can detect at least up to 300% strains and they provide a stable signal without hysteresis. The study is based on the premise that it is advantageous to have a fully fluidic composition in soft robots (i.e. sensors, actuators, logic, and power sources).

The authors mention that one of the drawbacks of current soft strain sensors is the use methods based on sealing fluids inside a closed silicone chamber which are inherently unstable for long-term measurements as leakages change the sensor output.

The main experiments for proprioception (Figure 1h) are meant to show how the sensor data can be used to infer information about the type of object being gripped. Unfortunately, for each different object, the data points show noticeable spreads (i.e. different $\Delta R/R$ values for gripping the same object). Even for the lemon, which has the majority of the data points clustered around a region in 3D space, there is one errant value closer with the lime data points. The authors should comment further on these results. Are the spreads due to variations on the relative positions between the gripper and objects for each experiment? If the positions/orientations were kept constant, what can be causing these variations?

If sensor measurements display a large variability, the author's claim that their information can be

used to infer the type of object being gripped is not strong.

The main experiment for exteroception (Movie 2, Fig. S3) is meant to show how the sensors can be connected directly to the actuation network and provide feedback. This result would be more impressive if the control speed was faster. Movie 2 (speed up 300%) and Fig. S3 show it takes ~ 1 minute for the actuation to be achieved! This is too slow for a robotic system and it is somewhat counter intuitive. Is the speed intentionally kept slow throughout the interaction? What is the speed of approach of the object? If not, what is the cause of the delay? Showing faster reaction times (i.e. higher bandwidths) will be more impressive.

The authors should clarify if the experiments to characterize the long-term stability and sensitivity of the sensors were done on a single sensor unit or if the results are average values of experiments using various units, this is not clear in the current text. Ideally, the results should be the average values of a few different samples/prototypes to demonstrate the robustness/repeatability of their design.

The overall results of the study would be more impressive if they could be benchmarked against other, similar-sized, soft sensors. For example, comparing the main results in Figure 1h and Figure 2, with results of similar-sized carbon grease or egain based strain sensors would be interesting. This would enable the authors to highlight the main advantages of the sensors.

As for many of these type of strain sensors, the authors should discuss design approaches to decouple measurements due to body deformation and physical contact. How will your controller determine if the feedback is indicative of one or the other?

Minor comments:

- In pages 3 and 9, provide modulus in SI units for your materials, not Shore values.
- In video 1, the sensor signals show some obvious oscillations during the period the fingers are "holding" payloads. The authors should explain the source of this phenomena. If the signals are filtered, this could indicate the presence of leaks.

Responses to Reviews of COMMSENG-21-0058-T

We thank the reviewers for the helpful suggestions and constructive comments. We have revised the text following the comments from the reviewers. Changes to the text are highlighted in the revised manuscript file in **yellow**. In the following, we address the reviewers' comments one by one.

Reviewer 1

Comment: *The paper claims on the development of a pneumatic strain sensor that can be easily fabricated and embedded into soft-bodied robots (hand). This paper is in line with recent attempt on electric-free designs for efficient implementation of sot robots with less burden from rigid components. Other evaluations on the operation of the sensor, and its application on equipping a soft robot hand with proprioceptive sensing were conducted.*

The good point in this design is that by only measuring the pressure, the strain or contact information can be extracted. Thus, no need to put anything into the soft body of the robot. However, the most unconvincing point is that it still needs pressure sensors for estimation of the strain based on equations 3-4. This reviewer expected something more than that, like pneumatic sensing+logic+actuation as a loop.

Response: We thank the reviewer for the excellent comments. As the reviewer pointed out, we have pressure sensors in the strain gauge system. However, the pressure sensors were mainly used to characterize the strain gauges. There is no intrinsic need to use pressure sensors with our strain gauges. Indeed, in Video 2 and Figure 4d we showed a simple robot where the actuators respond directly to the sensor signal, entirely without external pressure sensors. In that system, the strain gauge output signal drives the actuators in response to an external stimulus.

Comment: *With the current design, this reviewer thinks that the bandwidth of the strain sensor cannot be so high. In fact, more evaluation on dynamic change needs to be conducted, ...*

Response: In the original manuscript, the dynamic behavior was characterized only up to 0.25 Hz. There was a practical reason for it: the mechanical tester we use cannot produce very fast accelerations, which would be necessary to produce the 66.7% strains for our strain gauge with a length of 30 mm.

Mechanical testers usually have gear boxes to produce large forces rather than large accelerations.

However, we now redid the experiments and tried to push the frequency as high as we could. The highest frequency that we could reliably reach was 1 Hz. New cyclic strain experiments with frequency varying from 0.1 Hz up to 1 Hz are reported in Figure S8. Figure S8 shows that the sensor response does not significantly decay even at 1 Hz, and we are not quite certain if this decay is from the mechanical tester reaching its limits or is this a bandwidth limitation of the sensor. Nevertheless, this data shows that the bandwidth of the sensor is at least 1 Hz, if not more. 1 Hz is already close to the range necessary for tracking the fastest motions PneuNet actuators can produce (e.g. 50 ms reported in ¹). So, we believe that at least when PneuNet actuators are used, the bandwidth of the actuators will be the bottleneck, rather than the bandwidth of the sensors. We added the following text to the manuscript:

To test the bandwidth of our strain gauges, we repeated the 67% cyclic strain experiments with frequencies ranging from 0.1 Hz up to 1 Hz, which was the maximum frequency our characterization setup could reliably produce. The results are shown in Fig. S8. The sensor has a clear response even at 1 Hz, which shows that the bandwidth of the sensor is at least 1 Hz, if not more. Taking the results in Figs. 2d, S7 and S8 together, we conclude that the strain gauge has a stable response even in dynamic strain.

Figure S8. Strain gauge under 66.7% cyclic strain with an increasing frequency.

Comment: ..., and [dynamic response should be] compared to other soft sensors, such as EGaln-based ones.

Response: We did this comparison adding a new Table S1, where we compare the values from our sensors to the values reported in the literature. We compare the sensors in terms of their materials, fabrication methods, maximum strains, hysteresis and tested frequency. The Discussion section was reorganized to reflect the Table S1.

Table S1. Recently reported stretchable strain sensors and their characteristics. N.R. = Not Reported

Sensor Materials	Fabrication method	Sensor type	Maximum strain tested (%)	Hysteresis (%)	Gauge factor (GF)	Tested frequency (Hz)	Reference
Ionic solution/Liquid metal/Silicone elastomer	Elastomer casting/Channel filling	Resistive	100	N.R.	~3	N.R.	2
Metal nanowire/elastomer composite	Elastomer casting/nanowire coating	Resistive	150	N.R.	~81	N.R.	3
Ti ₃ C ₂ T _x -AgNW-based ink/polyurethane	Screen-printing	Resistive	83	20*	>200	1	4
Liquid metal/silicone elastomer	Elastomer casting/Channel filling	Resistive	550	N.R.	4.95	N.R.	5
Graphite and carbon black nanoparticles/cotton	Manual coating	Resistive	400	~0**	~100,000	0.2	6
Silver flake ink/thermoplastic polyurethane	Screen-printing	Resistive	2	17	5.8	N.R.	7
lonigel coped with Fe ₃ O ₄ particles	Synthesis of nanoparticles	Resistive	2000	~13***	20	0.005***	8
Ionic hydrogel/Silver nanofiber	Manual stacking	Capacitive	1000	~0****	165	0.08	9
Expanded intercalated graphite/fabric	Direct writing, screen-printing	Capacitive	250	~0	~0.5-1.2	0.75	10
Parylene/Gold film/VHB elastomer	Manual stacking	Capacitive	140	~0	~3	N.R.	11
Silicone elastomer	Elastomer casting	Pneumatic Resistance	300	~0*****	~1	1	This work

*Estimated from Fig. 5c in⁴

**Until 75%, then hysteresis observed

***Estimated from Fig. S11 in⁸

****100% strain, some hysteresis with 400% strain

*****Measured in 50%

Comment: *The authors mentioned about low hysteresis of the sensor, but it might due to the fact that it was not conducted at high frequency.*

Response: Testing at low frequencies was intentional. When we referred to hysteresis, we specifically intended to study *rate-independent hysteresis*, akin to hysteresis in relays or hysteresis in ferromagnetic materials. This is different from *rate-dependent hysteresis*, which the reviewer is referring to.

In our sensors, rate-dependent hysteresis could come from e.g., viscosity of the elastomer, and rate-independent hysteresis could come from e.g.,

plastic deformations. We specifically wanted to study the rate-independent effects in our experiments, and thus conducted the experiments at slow speeds.

To study the rate-dependent (viscous) hysteresis, we now redid the hysteresis experiment with 3 different speeds, ranging from 0.1 mm/s up to 1 mm/s, which was again the highest we could reliably produce with our machine. The results are shown in Fig. S6. No hysteresis can be observed even at the highest speeds we can produce with our sensor. However, notice that the pneumatic resistance data and the distance data are from two different devices and had to be manually aligned. So, the results only show that the approach and retraction curves have similar shapes, but they cannot be used to conclude if there is some phase lag between the mechanical tester and the sensor output. We revised the text as follows:

The **rate-independent** hysteretic behavior of the strain gauge was studied by recording $\Delta R/R_c$ when 50% engineering strain was applied for five cycles. In each cycle, the strain gauge was first stretched and then returned to its original length, at a constant strain rate. The strain rate was kept low (0.01 mm/s) to avoid rate-dependent effects e.g. dynamic viscoelastic effects. The results are shown in Fig. 2c. The **rate-independent** hysteresis, if there was any, was smaller than the noise level of the sensor (5% with the tested range). Even after averaging the data from the five cycles, no hysteresis was observed (Fig. 2c). Two other strain gauge samples with the same design were also tested and they showed similar non-hysteretic behavior (Fig. S5).

To study the rate-dependent hysteresis (e.g. viscous effects), we did the hysteresis experiment with three different speeds ranging from 0.1 mm/s up to 1 mm/s (Fig. S6). No hysteresis could be observed even at the highest speed. However, it should be noted that the pneumatic resistance data and the distance data are from two different devices and had to be manually aligned. Thus, the results only show that the approach and retraction curves have similar shapes, but they cannot be used to conclude if there is some phase lag between the mechanical tester and the sensor output.

Figure S5. Hysteresis measurements reported in Figure 2c repeated for two other samples with the same design.

Figure S6. Rate-dependent hysteresis of the strain gauge. The gauge was strained with a speed of **a** 0.1 mm/s, **b** 0.5 mm/s and **c** 1 mm/s.

Reviewer 3

Comment: *In this paper, the authors have created soft elastomer material with included microchannels. This fluidic sensing design is used in combination with a pneumatic setup to measure change in relative pneumatic pressure drops resulting in strain measurements.*

Though this is an interesting concept. However, there are a number of concerns:

1) The contribution of this paper is not clear, and the argumentation is not logical:

- In the introduction, the authors made a series of conclusions that are vague and, hence, incorrect. For example, it is stated that soft material robots "are simple to fabricate using elastomer casting, are fast, and achieve large deformations..." In my opinion, these statements would need to be revised and differentiated. For instance, there is increasing complexity in some soft robots, e.g., systems that have multiple molding steps. What does the author mean that soft robots are "fast"? Do they mean that the response is quick? This behavior is due to choosing appropriate pressure regulators, for instance.

Response: All authors thank the reviewer for their excellent suggestions. We agree that the start of the introduction was vague at the initial state. In the part of the introduction the reviewer pointed out vague arguments, the authors discuss about the advantages of the fluidic actuators compared to other soft actuators. We have revised the suggested statements as follows:

The four key components of robots are sensors, actuators, logic, and power source. In classic robots, these components are usually electrical. However, in soft robots¹²—robots made out of

soft materials—fluidic actuators^{7,13–16} have several advantages over other actuator types^{17,18} (e.g. shape memory alloys and cable-driven actuators): they can obtain high grasping forces,¹⁹ are fast to actuate,¹ and achieve large deformations²⁰. Breakthroughs in soft pneumatic actuators^{7,13,14} have inspired the use of fluidics for other soft robot components as well. Pneumatic circuits have been used for controlling soft robots,^{21–24} and gas from chemical reactions has been used as a pneumatic power source²⁵. This progress suggests that sensors in soft robots could also be entirely fluidic. However, soft fluidic sensors have received less attention than the other soft robot components.

Comment: *The review sections list a number of research efforts without describing the fundamental concepts. For instance, when looking into soft fluidic sensors, there is a common theme among these types of sensors.*

Response: The reviewer also pointed out that in the literature review section we describe many stretchable sensors without describing the fundamental concept. We agree that the section describing soft fluidic sensors should be more detailed and we revised the text as follows:

Soft fluidic sensors fabricated using the same materials as the rest of robot would solve all the aforementioned problems. In the literature, several fluidic sensors have been proposed that are based on measuring the pressure inside a sealed air chamber. When the shape of the chamber changes due to elongation, compression, or bending, its volume changes, resulting in a pressure change inside the chamber. For example, Yang et al.²⁶ proposed integrating a pneumatic chamber into a gripper for measuring contact force and curvature, Choi et al.²⁷ proposed a three-axis force sensor based on three radially symmetric pneumatic chambers, and Tawk et al.²⁸ proposed pneumatic sensing chambers for human-machine interfaces. Related to these ideas, Agaoglu et al.²⁹ proposed a solution with two chambers, one filled with liquid and another with air, with a channel in between. When the liquid chamber deforms, the water-air interface is displaced, which can be measured by using an image-based measurement.

Comment: *In fact, the authors conclude that these “methods based on sealing fluids inside a closed silicone chamber”. Further, it is “argue[d]” that these methods “are inherently unstable for long-term measurements”. Looking at the design and working principle of the authors’ work, this fundamental concept has also been adapted here measuring relative pressures. The main critic is that leakages might change the sensor output. If this is the disadvantage of current systems, how does this work overcome this limitation. More important, it would be good to understand how the sensor is affected by leakages.*

Response: We argued that the other fluidic sensors based on sealing the fluid inside the closed chamber are unstable for long term measurements. This is reported in the previous literature of pneumatic strain sensors.²⁷ We showed that our strain gauges can be used up to 12 hours. To study the effect of a leakage, we artificially induced leaks to the sensor and showed

that leaks only result in a gradual degradation of the performance of the sensor. Figure S4 shows the results. We revised the text as follows:

A leak in the strain gauge could change the strain gauge response. To study this, we measured the strain gauge response while artificially inducing leaks to the strain gauge. In the first experiment, we attached a valve in parallel with the strain gauge and opened the valve to varying degrees. Fig. S4a shows the results. When the valve is fully open, the strain gauge signal is fully grounded, and no signal is observed. However, as the valve is partially open, we can always measure some signal, albeit with a smaller signal-to-noise-ratio. This shows that leaks reduce mainly the signal amplitude, but this can be compensated with recalibration. In the second experiment, we punctured the strain gauge in different positions with 0.3 mm and 0.5 mm needles. Figure S4b shows the results. Fig. S4b shows that the strain gauge loses some of the signal amplitude as it starts to leak due to punctures, but again, the degradation of the signal is gradual as more punctures are added and at least in the case of minor leaks, this loss of signal can be compensated by recalibrating the strain gauge. Overall, these two experiments show that leaks do not result in catastrophic failure of the strain gauge, but rather just lower the signal-to-noise-ratio.

Figure S4. Behavior of pneumatic strain gauge in two types of artificial leaks: **a**, pneumatic valve was added in parallel with the strain gauge and it was opened with varying degrees. **b**, The strain gauge was punctured at different positions (inset). At each position, the strain gauge was first punctured with a 0.3 mm needle, and then with a 0.5 mm needle. After each puncture, the strain gauge response was tested for 5 cycles.

Comment: *The authors refer to the analogy to electric circuits. This analogy is well known. I am not sure what the reasoning for including this section is?*

Response: We deleted the general section about electrofluidic analogy. We now only mention electrofluidic analogy when starting to analyze our the strain gauge, to justify why we are using electrical symbols throughout the text.

Comment: *2) Some sections could benefit from an improved structure. For instance, the Results section begins with proof of concept, then reports on the results when integrating the sensor into a gripper. Later in the section, the authors report on stability and sensitivity. I advise to give a FULL characterization of the strain gauge sensor before reporting on performance criteria when embedding the sensor into a robotic gripper.*

3) The unstructured approach is also reflected in Figs 1 and 2. Both figures partially include characterization results of the sensor. I suggest changing these figures. One figure could include the entire overview, setup and results of the main concept. The other figure could then give an overview of the application and its results.

Response: We agree that the way suggested by the reviewer is a better way to structure the paper. We have now reorganized the paper so that Figures 1&2 report the strain gauge and its characterizations, while Figure 3 reports the application of the strain gauge as a curvature or tactile sensor and finally Figure 4 reports the integration of the strain gauges into a three-fingered robotic gripper.

Comment: *4) It is important to include figures that help to understand the work and that add scientific value. For instance, Fig. 1a does not add significant content. A setup of the real system is missing.*

Response: We agree that a photograph of the entire experimental setup would be helpful and added it to the supplementary material (Figure S12). Figure 1 has been redone to better communicate the sensor results but we did not quite agree that Figure 1a (1e in the revised manuscript) would not add any significant content, as it is the only figure where the real sensor and its channels can be clearly seen. Figure 1e demonstrates to the reader in a quick way how soft and stretchable our sensor is.

Figure S12. Measurement set up **a**, for the strain gauge and **b**, the soft pneumatic gripper.

Comment: *5) I suggest focusing the paper on the actual sensor and the characterization. The application of the sensor to a gripper, which includes a description of the manufacturing process, experiments etc., lacks depth (due to space limitation).*

Response: Thank you for the suggestion. As mentioned before, we have now rearranged the paper. The paper focuses first on the sensor, leaving the integration of the sensor into the soft robotic gripper last. We agree that the part describing the gripper fabrication and demonstrations were short. We revised the text as follows:

Gripper fabrication

The gripper was fabricated by connecting three pneumatic actuators together with a 3D printed piece (Clear resin, Form 2, Formlabs). Detailed drawings are provided in the supplementary material. The actuator design was same as used previously in the work, except here the strain gauges were integrated in the bottom part of the actuators (Fig. 4a and Fig. S13). The strain gauge R_4 used for exteroceptive measurements in the gripper palm had a channel with length of 221 mm and a width and a height of 500 μm . The channel cross-section was increased to reduce the pneumatic resistance of the palm sensor, to make the gripper close faster. The palm sensor was attached to the 3D printed connecting piece with a tape. The pneumatic connectors to the actuators and strain gauges were provided through the 3D printed connecting piece (Fig. S14).

Gripper demonstrations

The grippers actuators (C_{1-3}) were connected in parallel with the palm sensor R_4 . They were all connected in series with the constant pneumatic restrictor R_{c4} (Teflon tube, length = 1591 mm, and diameter 0.5 mm) which was connected to the constant pneumatic source $P_{\text{supply}2}$. The strain sensors in the gripper's fingers were all connected to the constant restrictors ($R_{c1,c2,c3}$) and then to constant pneumatic source $P_{\text{supply}1}$.

In the object recognition experiment, the 3D printed connecting piece was attached to the vertical motor (TA.XT Plus, Stable Microsystems) with a screw (Fig. 4c photographs and a full measurement setup in Fig. S12b). The pressure was measured as described in the section *Strain measurements*. The picked objects were placed in the same position under the gripper manually, and the gripper was moved in contact with the vertical motor (Video 1).

In the self-closing gripper experiment, the gripper was placed with fingers pointing up (Fig. 4d and Video 2). Then a heavy object was placed onto the palm sensor (R_4) by hand, causing the gripper fingers to close. Removing the object caused the fingers to open again.

Comment: 6) *The Discussions section is vague. For instance, the authors respectively claim that the sensor is inexpensive, simple and “trivial” to integrate. There are no proofs for these claims in the paper. The section does not reflect on the contributions.*

Response: The reviewer has excellent points, and we agree that some of the claims were vague. We revised these claims as:

- 1) The sensor does not significantly increase the amount of material consumed in making a soft robot, as the sensor is significantly smaller than typical pneumatic network actuators, and the sensor is fabricated using the exact same materials as the actuators.
- 2) The sensors are simpler to fabricate than sensors based on liquid metals, as both sensor types require a microfluidic channel, but our sensor does not need filling of the microchannels with anything else than air.
- 3) Compared to electrical resistive sensors, our pneumatic sensors have much less hysteresis, which is shown in the new Table S1.

We revised the conclusion as follows:

Our sensors are made of soft silicone elastomer, the same material that is commonly used to make the bodies of soft robots, so it can be fabricated using similar materials and processes as the rest of the robot. The sensors do not significantly increase the amount of material consumed in making a pneumatically actuated soft robot, as the sensor is significantly smaller than typical pneumatic network actuators. Pneumatic strain gauges do not require any other materials, such as ionic conductors,³⁰ liquid metals,³¹ or inks,^{7,10} which is a clear advantage of the proposed approach (Table S1). Apart from air and silicone interfaces, heterogeneous materials are not used in our sensors. This is beneficial since delamination often occurs at heterogeneous material interfaces.

[...]

To conclude, the proposed soft pneumatic strain gauges offer a way to integrate proprioceptive and exteroceptive sensing into pneumatic soft robots. Compared to the well-studied liquid alloy

based resistive sensors, our pneumatic sensors have less fabrication steps, as both sensor types require a microfluidic channel, but our sensor does not need the channel to be filled with anything else than air. Compared to electrical resistive sensors, our pneumatic sensors have much less hysteresis and drift. Our sensors also operate on the same pneumatic principles as the soft pneumatic actuators so only one power system is needed. In future, we expect our sensors to see more practical applications in fully pneumatic, electronics-free soft robotic devices.

Comment: *In fact, the authors finally conclude that “This paves the way toward fully pneumatic, electronics-free soft robots” – the title of the paper suggests that this has already been achieved in the paper.*

Response: The sentence was removed.

Reviewer 4

Comment: *The authors present a study on pneumatic strain sensors based on meandering microchannels embedded in a soft elastomer material. The sensors can detect at least up to 300% strains and they provide a stable signal without hysteresis. The study is based on the premise that it is advantageous to have a fully fluidic composition in soft robots (i.e. sensors, actuators, logic, and power sources).*

The authors mention that one of the drawbacks of current soft strain sensors is the use methods based on sealing fluids inside a closed silicone chamber which are inherently unstable for long-term measurements as leakages change the sensor output.

The main experiments for proprioception (Figure 1h) are meant to show how the sensor data can be used to infer information about the type of object being gripped. Unfortunately, for each different object, the data points show noticeable spreads (i.e. different $\Delta R/R$ values for gripping the same object). Even for the lemon, which has the majority of the data points clustered around a region in 3D space, there is one errant value closer with the lime data points. The authors should comment further on these results. Are the spreads due to variations on the relative positions between the gripper and objects for each experiment? If the positions/orientations were kept constant, what can be causing these variations?

If sensor measurements display a large variability, the author’s claim that their information can be used to infer the type of object being gripped is not strong.

Response: We thank the reviewer for all the excellent comments. We agree that the data that was previously in Figure 1h (Figure 4c in the revised paper) had noticeable spreads. We strongly believe that the spread was due to real variations how the gripper got grasp of the object and not an artifact from

the sensor, and we believe that the one outlier may have been due to the object slipping from the grasp of the gripper. However, unfortunately, we previously did not record video evidence from every gripping experiment, which could have revealed if there was any slippage.

Thus, we redid all the experiments in the new Fig. 4c and Video 1, while recording also video evidence from each gripping experiment. In the revised data, there is no more clear outliers, and picking the bottle shows more spread than the picking of the fruits. The video evidence shows clearly that the gripper picks the bottle slightly in a different way each time (Fig. S9), while there are less variations in how the fingers bend when the gripper picks the lime and the lemon (Fig. S9). These results are strongly indicative that the variations in the sensor signals are in large part due to real variations how the gripper fingers bend during gripping. We revised the text as follows:

Figure S9. Snapshots from each picking experiment (1-10) of the water bottle, the lime, and the lemon in Figure 4c.

When the gripper fingers were closed, the curvature of the fingers depended on the shape of the target object. The shape and size of the gripped object allowed the fingers to bend by different amounts, and this difference can be observed from the data. To determine whether we could infer the type of the object from the sensor signals alone, we selected three differently shaped objects, a lemon (183 g), a lime (91 g), and a water bottle (24 g), as shown in Video 1 and Fig. 4c. Each object was gripped ten times, and data were captured from each strain gauge. The object was changed after every two picking experiments to avoid systemic drift and carry over effects. From each experiment, we calculated the change in sensor signal before and after gripping the object. Fig. 4c shows a 3D scatter plot with each axis corresponding to a different sensor. The data from the water bottle picking is the most scattered. Snapshots of videos recorded from the picking experiments (Fig. S9) show that the fingers bend slightly differently when picking the water bottle, because of its odd shape, while there are no noticeable differences in how the lime and lemon are grasped, because their shape is more regular. Thus, these results are strongly indicative that the variations in the sensor signals are in large part due to real variations how the gripper fingers bend during gripping. In conclusion, the objects formed clear clusters, suggesting that the sensor data can be used to infer information about the type of the gripped object.

Fig 4. Fully pneumatic soft robot with perception. **a**, Schematic of pneumatic gripper with perception: fingers including proprioceptive sensors and the palm includes an exteroceptive sensor. **b**, Electrical equivalent circuit for the gripper: actuators can be considered as capacitors $C_{1,2,3}$, constant pneumatic restrictors as resistors $R_{c1,c2,c3,c4}$, and pneumatic strain gauges as variable resistors $R_{1,2,3,4}$. **c**, $\Delta R/R_c$ for each pneumatic strain gauge when three different objects (a water bottle **(I)**, a lime **(II)** and a lemon **(III)**), are each gripped ten times. **d**, Movement of three fingers when pressure is applied on the gripper palm: fully open **(I)**, fully closed **(II)**, opening **(III)** and fully open **(IV)**. Scale bars 2 cm.

Comment: *The main experiment for exteroception (Movie 2, Fig. S3) is meant to show how the sensors can be connected directly to the actuation network and provide feedback. This result would be more impressive if the control speed was faster. Movie 2 (speed up 300%) and Fig. S3 show it takes ~ 1*

minute for the actuation to be achieved! This is too slow for a robotic system, and it is somewhat counter intuitive. Is the speed intentionally kept slow throughout the interaction? What is the speed of approach of the object? If not, what is the cause of the delay? Showing faster reaction times (i.e., higher bandwidths) will be more impressive.

Response: The speed was not intentionally kept low, but rather an unfortunate consequence of trying to keep the sensor dimensions consistent throughout the paper: the channel cross-section was $200\ \mu\text{m} \times 200\ \mu\text{m}$. The self-closing gripper can be understood as a RC -circuit, where the sensor resistance affects R , and the capacitance C is from the pneumatic chambers of the actuators and from the tubing. The time constant τ of an RC -circuit is:

$$\tau = RC$$

This suggests an easy way to make the gripper close faster: by reducing R , the response time can be decreased. The resistance can be decreased by increasing the channel cross-section. This allows the response time to be tailored to the needs of the targeted application. Note that faster is not always better, as it also makes the gripper likely to close also in response to spurious signals.

We redid the Video 2, Fig. S10 and Fig. 4d snapshots using a sensor with a larger channel cross-section of $500\ \mu\text{m} \times 500\ \mu\text{m}$. With this new sensor, the gripper closes in $\sim 8\ \text{s}$ (see New Video 2 and snapshots in Figure 4d). This is 13% of the original time it took for the gripper to close. We revised the text as follows:

To demonstrate routing sensor signals directly to the actuators of a soft robot, we placed one strain gauge (channel width and height: $500\ \mu\text{m}$) onto the gripper palm (Fig. 4a) and connected that palm sensor (R_4 in Fig. 4b) in parallel with the pneumatic fingers (C_{1-3} in Fig. 4b). When R_4 increases, the pressure P_4 inside the fingers is to increase. The response of the pneumatic gripper to external compression of the strain sensor was studied by pressing the strain gauge with a solid object (Fig. 4d and Video 2). The results show that the compression indeed increases P_4 (Fig. S10), causing the fingers to bend. When the external compression is released, the fingers return to their initial shape. The closing of the fingers takes $\sim 8\ \text{s}$ which is caused by the pneumatic resistance and capacitance in the gripper fingers, sensor R_4 and tubing. The self-closing gripper can be understood as an RC -circuit, where the sensor resistance affects R , and the capacitance C is from the pneumatic chambers of the actuators and from the tubing. The time constant τ of an RC -circuit is given by $\tau = RC$, which suggests that the response time of the gripper can be adjusted by changing the resistance i.e. the length and cross-section of the meandering strain gauge channel. This was also showed experimentally by using a strain gauge with a smaller microchannel (channel width and height: $200\ \mu\text{m}$). It then took $\sim 1\ \text{min}$ for the gripper to close. To conclude, this is a simple example of an all-pneumatic system that responds to external stimulus by changing its shape, and the response can be tailored to the needs of the application by changing the sensor design. Taking these results together with the results in Fig. 3e&f, the

proposed sensors can be used as soft tactile pressure sensors for measuring the external forces in soft robot applications.

Comment: *The authors should clarify if the experiments to characterize the long-term stability and sensitivity of the sensors were done on a single sensor unit or if the results are average values of experiments using various units, this is not clear in the current text. Ideally, the results should be the average values of a few different samples/prototypes to demonstrate the robustness/repeatability of their design.*

Response: In Figure 2b, the sensitivity (gauge factor) was already calculated by taking the average of the five different strain gauge samples with the same design.

In Figures 2a&c&d, we indeed only reported the results from a single sample. The reason for this was that averaging time-dependent signals carries the risk of losing, hiding or even giving misleading information. For example: time-averaging two sinusoidal signals, where one has a slight phase-shift relative to the other might result in a completely flat signal. Similarly, averaging two sinusoidal signals with slightly different frequencies could result in constructive and destructive interference, as the two signals go into phase and out of phase.

We still agree with the reviewer that the measurements in Figures 2a&c&d should be repeated for multiple samples. We have now repeated the characterizations for two other samples and report these measurements in the new Supplementary Figures S3, S5 and S7, respectively. The results from the other samples are very comparable to the results from the sample reported in the main text, showing that different samples behave quite similarly.

We also revised the text to make it clear how many samples were used and to reflect the measurements done with other samples:

To characterize the long-term stability of the fabricated soft pneumatic strain gauge, we recorded $\Delta R/R_c$ while maintaining a gauge at 50% engineering strain for 12 hours (Fig. 2a). The value of $\Delta R/R_c$ drifted from 0.127 to 0.119, with the relative change being 5.9%. After the release, $\Delta R/R_c$ returned almost to the initial value, and the strain gauge did not show significant overshooting behavior. Other two strain gauge samples with same design were also tested (Fig. S3), and all three strain gauges had similar stable response over time. Overall, the long-term response of the strain gauge was reliable and reversible.

A leak in the strain gauge could change the strain gauge response. To study this, we measured the strain gauge response while artificially inducing leaks to the strain gauge. In the first experiment, we attached a valve in parallel with the strain gauge and opened the valve to varying degrees. Fig. S4a shows the results. When the valve is fully open, the strain gauge signal is fully

grounded, and no signal is observed. However, as the valve is partially open, we can always measure some signal, albeit with a smaller signal-to-noise-ratio. This shows that leaks reduce mainly the signal amplitude, but this can be compensated with recalibration. In the second experiment, we punctured the strain gauge in different positions with 0.3 mm and 0.5 mm needles. Figure S4b shows the results. Fig. S4b shows that the strain gauge loses some of the signal amplitude as it starts to leak due to punctures, but again, the degradation of the signal is gradual as more punctures are added and at least in the case of minor leaks, this loss of signal can be compensated by recalibrating the strain gauge. Overall, these two experiments show that leaks do not result in catastrophic failure of the strain gauge, but rather just lower the signal-to-noise-ratio.

The sensitivity of a strain gauge can be studied by its gauge factor (GF), which is the relative change in the output signal to the applied strain:

$$GF = \frac{dR}{R} \cdot \frac{L}{dL} \quad (5)$$

Due to our measuring configuration, we could not measure R , only R/R_c . Additionally, we wanted to study the gauge factor as a function of the engineering strain ε ($dL = L_0 d\varepsilon$). By substituting these variables into Eq. (5), we obtained

$$GF = \frac{d(R/R_c)}{d\varepsilon} \cdot \frac{R_c}{L_0} \cdot \frac{L}{R} = \frac{d(R/R_c)}{d\varepsilon} \cdot \frac{1+\varepsilon}{R/R_c}, \quad (6)$$

which is shown in Fig. 2b. The GF is calculated by using an average R/R_c of the five strain gauge samples (dashed blue line in Figure 2b). The GF varied from 0.71 to 1.2 but was very small in general. Doubling the length roughly doubled the resistance.

The rate-independent hysteretic behavior of the strain gauge was studied by recording $\Delta R/R_c$ when 50% engineering strain was applied for five cycles. In each cycle, the strain gauge was first stretched and then returned to its original length, at a constant strain rate. The strain rate was kept low (0.01 mm/s) to avoid rate-dependent effects e.g. dynamic viscoelastic effects. The results are shown in Fig. 2c. The rate-independent hysteresis, if there was any, was smaller than the noise level of the sensor (5% with the tested range). Even after averaging the data from the five cycles, no hysteresis was observed (Fig. 2c). Two other strain gauge samples with the same design were also tested and they showed similar non-hysteretic behavior (Fig. S5).

To study the rate-dependent hysteresis (e.g. viscous effects), we did the hysteresis experiment with three different speeds ranging from 0.1 mm/s up to 1 mm/s (Fig. S6). No hysteresis could be observed even at the highest speed. However, it should be noted that the pneumatic resistance data and the distance data are from two different devices and had to be manually aligned. Thus, the results only show that the approach and retraction curves have similar shapes, but they cannot be used to conclude if there is some phase lag between the mechanical tester and the sensor output.

To investigate whether the strain gauge is stable under dynamic strain, we recorded $\Delta R/R_c$ when 67% engineering strain was applied for 1000 cycles with a frequency of 0.25 Hz (Fig. 2d). Some

minor variations were observed from one cycle to another (Fig. 2d), but overall, the response was very stable for the duration of the whole experiment, which was over an hour. To confirm that these results were reproducible, we repeated the cyclic loading experiment with two additional samples. The results are shown in Figure S7 and are very close to the results in Figure 2d.

To test the bandwidth of our strain gauges, we repeated the 67% cyclic strain experiments with frequencies ranging from 0.1 Hz up to 1 Hz, which was the maximum frequency our characterization setup could reliably produce. The results are shown in Fig. S8. The sensor has a clear response even at 1 Hz, which shows that the bandwidth of the sensor is at least 1 Hz, if not more. Taking the results in Figs. 2d, S7 and S8 together, we conclude that the strain gauge has a stable response even in dynamic strain.

Fig. 2. Stability and sensitivity of the soft pneumatic strain gauges. **a**, Stability of the strain gauge when a static 50% engineering strain is held for 12 h and photographs from the measurements, scale bar: 1 cm. **b**, Gauge factor of the pneumatic strain gauge. The dashed line is the mean R/R_c of the five tested samples shown in Fig. 1e, with the shaded area showing standard deviation. The solid pink line is the gauge factor (average of five strain gauges), calculated using Eq. (6). **c**, Hysteresis of the strain gauge when 50% engineering strain is applied for five cycles (strain rate: 0.01 mm/s). The strain gauge shows no hysteretic behavior. **d**, Stability of the strain gauge when a cyclic 67% engineering strain is applied for 1000 cycles (frequency: 0.25 Hz).

Figure S3. Long term stability measurements reported in Figure 2a repeated for two other samples with the same design.

Figure S5. Hysteresis measurements reported in Figure 2c repeated for two other samples with the same design.

Figure S6. Rate-dependent hysteresis of the strain gauge. The gauge was strained with a speed of **a** 0.1 mm/s, **b** 0.5 mm/s and **c** 1 mm/s.

Figure S7. Cyclic strain measurements reported in Figure 2d repeated for two other samples with the same design.

Figure S8. Strain gauge under 66.7% cyclic strain with an increasing frequency.

Comment: *The overall results of the study would be more impressive if they could be benchmarked against other, similar-sized, soft sensors. For example, comparing the main results in Figure 1h and Figure 2, with results of similar-sized carbon grease or e-gain based strain sensors would be interesting. This would enable the authors to highlight the main advantages of the sensors.*

Response: We agree that a better comparison to existing literature should have been done. We have now added a new Supplementary Table 1, comparing our results to the different sensors reported in the literature.

Comment: *As for many of these type of strain sensors, the authors should discuss design approaches to decouple measurements due to body deformation and physical contact. How will your controller determine if the feedback is indicative of one or the other?*

Response: We agree that the decoupling of body deformations (proprioception) from physical contact (exteroception) is important challenge with all types of soft stretchable sensors, not only ours. All solutions to this problem are based on the fact that sensors placed in different locations (at the surface of the robot vs. buried deep inside the robot) respond in different degrees to body deformations and physical contact. Thus, with multiple sensors, placed in different locations, different stimuli can be tested and their contribution to different sensor signals observed. This gives the mapping from the different stimuli to the sensor signals; the mapping can then be inverted to decouple the different stimuli from the observed sensor signals. The decoupling algorithm can be as simple as a linear matrix,³² but also machine learning algorithms have been proposed.³³ We revised the discussion as follows:

We have shown that our sensors respond not only to linear strains but also to transverse compression and bending. In soft robotic applications, the strain gauge could respond to multiple types of stimuli, including body deformations (proprioception) and physical contact (exteroception), and thus the contribution of different stimuli needs to be decoupled. All solutions to this problem are based on the fact that sensors placed in different locations—at the surface of the robot vs. buried deep inside the robot—respond in different degrees to body deformations and to physical contact. Thus, with multiple sensors, placed at different locations, different stimuli can be tested and their contribution to different sensor signals observed. This gives a mapping from the different stimuli to the sensor signals; the mapping can then be inverted to decouple the different stimuli from the observed sensor signals. In the literature, signals from different types of sensors have been decoupled using a linear mapping,³² but also machine learning algorithms have been used.³³

Comment: *Minor comments:*

- In pages 3 and 9, provide modulus in SI units for your materials, not Shore values.

Response: We added 100% (tensile strain) modulus values for the elastomers we used, as these are usually reported in the datasheets of soft silicone elastomers.

Comment: - In video 1, the sensor signals show some obvious oscillations during the period the fingers are “holding” payloads. The authors should explain the source of this phenomena. If the signals are filtered, this could indicate the presence of leaks.

Response: The data for the Video 1 was rerecorded because the reviewer pointed out that the data had noticeable spreads and we wanted to find out the reason for those spreads. In the new Video 1 such oscillations cannot be seen. All pressure data is filtered with a first-order low-pass filter with a cut-off frequency of 1 Hz.

We are not comfortable speculating what might have been the source of the earlier disturbances observed in the data; whether it was leaks or some external disturbance source (e.g., pressure fluctuations or mechanical disturbances). Nevertheless, even there was leaks, we have now shown that leaks do not prevent the pneumatic strain gauge from working; they only reduce its signal to noise ratio. As described earlier (comments from Reviewer 3), we performed an experiment where we artificially induced leaks to the sensor and showed that leaks only result in a gradual degradation of the performance of the sensor. We revised the text as follows:

A leak in the strain gauge could change the strain gauge response. To study this, we measured the strain gauge response while artificially inducing leaks to the strain gauge. In the first

experiment, we attached a valve in parallel with the strain gauge and opened the valve to varying degrees. Fig. S4a shows the results. When the valve is fully open, the strain gauge signal is fully grounded, and no signal is observed. However, as the valve is partially open, we can always measure some signal, albeit with a smaller signal-to-noise-ratio. This shows that leaks reduce mainly the signal amplitude, but this can be compensated with recalibration. In the second experiment, we punctured the strain gauge in different positions with 0.3 mm and 0.5 mm needles. Figure S4b shows the results. Fig. S4b shows that the strain gauge loses some of the signal amplitude as it starts to leak due to punctures, but again, the degradation of the signal is gradual as more punctures are added and at least in the case of minor leaks, this loss of signal can be compensated by recalibrating the strain gauge. Overall, these two experiments show that leaks do not result in catastrophic failure of the strain gauge, but rather just lower the signal-to-noise-ratio.

Figure S4. Behavior of pneumatic strain gauge in two types of artificial leaks: **a**, pneumatic valve was added in parallel with the strain gauge and it was opened with varying degrees. **b**, The strain gauge was punctured at different positions (inset). At each position, the strain gauge was first punctured with a 0.3 mm needle, and then with a 0.5 mm needle. After each puncture, the strain gauge response was tested for 5 cycles.

References:

1. Mosadegh, B. *et al.* Pneumatic networks for soft robotics that actuate rapidly. *Adv. Funct. Mater.* **24**, 2163–2170 (2014).
2. Chossat, J. B., Park, Y. L., Wood, R. J. & Duchaine, V. A soft strain sensor based on ionic and metal liquids. *IEEE Sens. J.* **13**, 3405–3414 (2013).
3. Kim, K.-H. *et al.* Highly Sensitive and Stretchable Resistive Strain Sensors Based on Microstructured Metal Nanowire/Elastomer Composite Films. *Small* **14**, 1704232 (2018).
4. Shi, X. *et al.* Bioinspired Ultrasensitive and Stretchable MXene-Based Strain Sensor via Nacre-Mimetic Microscale ‘Brick-and-Mortar’ Architecture. (2018) doi:10.1021/acsnano.8b07805.
5. Gao, Q. *et al.* Microchannel structural Design For a Room-temperature Liquid Metal Based super-stretchable sensor. *Sci. Rep.* **9**, (2019).
6. Souri, H. & Bhattacharyya, D. Highly sensitive, stretchable and wearable strain sensors using fragmented conductive cotton fabric. *J. Mater. Chem. C* **6**, 10524–10531 (2018).
7. Koivikko, A., Raei, E. S., Mosallaei, M., Mantysalo, M. & Sariola, V. Screen-printed curvature sensors for soft robots. *IEEE Sens. J.* **18**, 223–230 (2017).
8. Mei Zhang, L. *et al.* Self-Healing, Adhesive, and Highly Stretchable Ionogel as a Strain Sensor for Extremely Large Deformation. *Small* **15**, 1804651 (2019).
9. Xu, H. *et al.* An ultra-stretchable, highly sensitive and biocompatible capacitive strain sensor from an ionic nanocomposite for on-skin monitoring. *Nanoscale* **11**, 1570–1578 (2019).
10. White, E. L., Yuen, M. C., Case, J. C. & Kramer, R. K. Low-Cost, Facile, and Scalable Manufacturing of Capacitive Sensors for Soft Systems. *Adv. Mater. Technol.* **2**, 1700072 (2017).
11. Nur, R. *et al.* A Highly Sensitive Capacitive-type Strain Sensor Using Wrinkled Ultrathin Gold Films. *Nano Lett.* **18**, 5610–5617 (2018).
12. Rus, D. & Tolley, M. T. Design, fabrication and control of soft robots. *Nature* **521**, 467–475 (2015).

13. Shepherd, R. F. *et al.* Multigait soft robot. *Proc. Natl. Acad. Sci. U. S. A.* **108**, 20400–20403 (2011).
14. Martinez, R. V. *et al.* Robotic tentacles with three-dimensional mobility based on flexible elastomers. *Adv. Mater.* **25**, 205–212 (2013).
15. Koivikko, A., Drotlef, D. M., Sitti, M. & Sariola, V. Magnetically switchable soft suction grippers. *Extrem. Mech. Lett.* **44**, 101263 (2021).
16. Katzschmann, R. K., Marchese, A. D. & Rus, D. Hydraulic autonomous soft robotic fish for 3D swimming. *Springer Tracts Adv. Robot.* **109**, 405–420 (2016).
17. Mazzolai, B. *et al.* Octopus-Inspired Soft Arm with Suction Cups for Enhanced Grasping Tasks in Confined Environments. *Adv. Intell. Syst.* **1**, 1900041 (2019).
18. Shamsavari, H. *et al.* Bioinspired underwater locomotion of light-driven liquid crystal gels. *Proc. Natl. Acad. Sci. U. S. A.* **117**, 5125–5133 (2020).
19. Zaidi, S., Maselli, M., Laschi, C. & Cianchetti, M. Actuation Technologies for Soft Robot Grippers and Manipulators: A Review. *Curr. Robot. Reports* **2**, 355–369 (2021).
20. Hawkes, E. W., Blumenschein, L. H., Greer, J. D. & Okamura, A. M. A soft robot that navigates its environment through growth. *Sci. Robot.* **2**, 1–8 (2017).
21. Drotman, D., Jadhav, S., Sharp, D., Chan, C. & Tolley, M. T. Electronics-free pneumatic circuits for controlling soft-legged robots. *Sci. Robot.* **6**, (2021).
22. Preston, D. J. *et al.* A soft ring oscillator. *Sci. Robot.* **4**, 1–10 (2019).
23. Preston, D. J. *et al.* Digital logic for soft devices. *Proc. Natl. Acad. Sci. U. S. A.* **116**, 7750–7759 (2019).
24. Song, S., Joshi, S. & Paik, J. CMOS-Inspired Complementary Fluidic Circuits for Soft Robots. *Adv. Sci.* (2021) doi:10.1002/advs.202100924.
25. Barlett, N. W. *et al.* A 3D-printed, functionally graded soft robot powered by combustion. *Science (80-.)*. **349**, 161–166 (2015).
26. Yang, H., Chen, Y., Sun, Y. & Hao, L. A novel pneumatic soft sensor for measuring contact force and curvature of a soft gripper. *Sensors Actuators A Phys.* **266**, 318–327 (2017).
27. Choi, H. & Kong, K. A Soft Three-Axis Force Sensor Based on Radially Symmetric Pneumatic Chambers. *IEEE Sens. J.* **19**, 5229–5238 (2019).
28. Tawk, C., in het Panhuis, M., Spinks, G. M. & Alici, G. Soft Pneumatic Sensing Chambers for Generic and Interactive Human–Machine Interfaces. *Adv. Intell. Syst.* **1**, 1900002 (2019).
29. Agaoglu, S. *et al.* Ultra-sensitive microfluidic wearable strain sensor for intraocular pressure monitoring. *Lab Chip* **18**, 3471–3483 (2018).
30. Truby, R. L. *et al.* Soft Somatosensitive Actuators via Embedded 3D Printing. *Adv.*

Mater. **30**, 1–8 (2018).

31. Larson, C. *et al.* Highly Stretchable Electroluminescent Skin for Optical Signaling and Tactile Sensing. *Science* (80-.). **351**, 1071–1074 (2016).
32. Zhao, H., O'Brien, K., Li, S. & Shepherd, R. F. Optoelectronically innervated soft prosthetic hand via stretchable optical waveguides. *Sci. Robot.* **7529**, eaai7529 (2016).
33. Van Meerbeek, I. M., De Sa, C. M. & Shepherd, R. F. Soft optoelectronic sensory foams with proprioception. *Sci. Robot.* **3**, 1–8 (2018).

REVIEWERS' COMMENTS:

Reviewer #1 (Remarks to the Author):

The authors showed great commitment by thoroughly revising the paper based on recommendation from reviewers. The manuscript is now ready for publication.

Reviewer #4 (Remarks to the Author):

The authors have addressed all of our comments satisfactorily. In particular, we really appreciate the updates in the figures, the extra new stability and sensitivity results, and the addition of Table 1 which will help the readers a quick comparison with the existing literature. We recommend the article for publication.

Responses to Reviews of COMMS-21-0058-T

We thank the reviewers for the helpful suggestions and constructive comments. We have revised the text following the comments from the reviewers. In the following, we address the reviewers' comments one by one.

Reviewer 1

Comment: *The authors showed great commitment by thoroughly revising the paper based on recommendation from reviewers. The manuscript is now ready for publication.*

Response: Thank you for the comment.

Reviewer 4

Comment: *The authors have addressed all of our comments satisfactorily. In particular, we really appreciate the updates in the figures, the extra new stability and sensitivity results, and the addition of Table 1 which will help the readers a quick comparison with the existing literature. We recommend the article for publication.*

Response: Thank you for the comment.